# Improving fragment-based ab initio protein structure assembly using low-accuracy contact-map predictions

S. M. Mortuza[1,3], Wei Zheng [1,3], Chengxin Zhang [1], Yang Li[1], Robin Pearce[1] & Yang Zhang [1,2✉]

Sequence-based contact prediction has shown considerable promise in assisting non-homologous structure modeling, but it often requires many homologous sequences and a sufficient number of correct contacts to achieve correct folds. Here, we developed a method, C-QUARK, that integrates multiple deep-learning and coevolution-based contact-maps to guide the replica-exchange Monte Carlo fragment assembly simulations. The method was tested on 247 non-redundant proteins, where C-QUARK could fold 75% of the cases with TM-scores (template-modeling scores) ≥0.5, which was 2.6 times more than that achieved by QUARK. For the 59 cases that had either low contact accuracy or few homologous sequences, C-QUARK correctly folded 6 times more proteins than other contact-based folding methods. C-QUARK was also tested on 64 free-modeling targets from the 13th CASP (critical assessment of protein structure prediction) experiment and had an average GDT_TS (global distance test) score that was 5% higher than the best CASP predictors. These data demonstrate, in a robust manner, the progress in modeling non-homologous protein structures using low-accuracy and sparse contact-map predictions.

---

[1] Department of Computational Medicine and Bioinformatics, University of Michigan, Ann Arbor, MI, USA. [2] Department of Biological Chemistry, University of Michigan, Ann Arbor, MI, USA. [3] These authors contributed equally: S. M. Mortuza, Wei Zheng. ✉email: zhng@umich.edu

A b initio protein structure prediction, which generally refers to approaches that model protein structures without using homologous templates in the PDB (Protein Data Bank), has attracted constant interest over the last several decades[1–7]. Consequently, considerable progress has been witnessed along this direction by the community-wide CASP (critical assessment of protein structure prediction) experiments[8–12]. For instance, while the success of ab initio modeling was limited to folding small proteins with lengths below 100 residues until only a decade ago[13–15], several advanced pipelines, including Rosetta[3] and QUARK[5], generated correct folds for challenging targets with lengths above 100 residues in the recent CASP experiments[16,17]. The advancements are primarily due to the development of advanced energy force fields and efficient search engines that help obtain the global energy minimum near the native state during the folding simulations. Nevertheless, the current force fields and search engines often fail to capture precise long-range atomic interactions in proteins. As a result, the modeling accuracy for large proteins with complicated topologies based on ab initio folding approaches has been inconsistent and still far from satisfactory[11,18].

One of the efficient ways to overcome the limitations in ab initio modeling is to incorporate long-range contacts, i.e., spatial adjacency of residue pairs with large sequence separations but that are close to each other in the three-dimensional (3D) structures, as restraints in the folding simulations. While the a priori knowledge of inter-residue contacts helps constrain the conformational search towards near-native states, sufficiently high accuracy contact prediction is required so that the modeling accuracy is not hindered because of too many falsely predicted contacts. Early efforts in contact prediction focused on coevolution[19] and machine learning[20,21], but the impact on ab initio structure folding was modest due to the limited accuracy of the contact-map prediction[22,23]. A leap in contact prediction accuracy was recently brought about by the introduction of direct coupling analysis (DCA)[24–27] and deep neural-network learning[28,29] techniques. While DCA helps remove translational contact noises from multiple sequence alignments (MSAs), supervised deep-learning techniques learn inherent contact patterns from PDB structures starting from co-evolutionary features derived from MSAs. Despite the remarkable progress in contact prediction, the success of current ab initio modeling protocols cannot be attained to their full potentials unless the predicted contacts are effectively integrated with the folding simulations. In particular, when the number of homologous sequences and therefore the accuracy of sequence-based contact prediction is low, how to balance the noisy contact-maps with the advanced folding simulation force fields to construct correct ab initio structure folds remains an important and challenging problem.

In this study, we developed a contact-guided ab initio folding program, C-QUARK, as an extension of QUARK[5,30], which has been ranked as one of the top methods in the CASP experiments since 2010[8–11]. To systematically explore the capacity of contact-map prediction, especially those with low accuracy, to improve ab initio folding, a 3-gradient (3 G) contact potential, characterized by three smooth platforms that account for both short- and long-distance gradients, is proposed and carefully tailored to incorporate the contact restraints with the QUARK-based folding simulations. The pipeline was rigorously benchmarked in comparison to QUARK, as well as other state-of-the-art structure modeling methods, on both CASP targets and a separate large-scale test dataset, where C-QUARK showed a remarkable advantage for modeling distant- and non-homologous targets. The results demonstrate, in a robust manner, the critical importance of a balanced combination of multiple complementary contact restraints with an advanced knowledge-based force field for improving the accuracy of ab initio protein structure prediction, especially for targets with complicated folding topologies.

It should be noted that after the work was completed, the field witnessed important progress brought by the introduction of distance[6,31] and inter-residue orientation[7] predictions integrated with quick gradient descent optimization. Nevertheless, given the special role of contact-map prediction in protein folding and the fact that most of the predicted distances and orientations are on residue pairs within short distances of each other (i.e., in contact), we believe it is of critical importance to study and benchmark separately the impact of contact-maps on the problem of ab initio protein structure prediction, and systematically examine the critical weaknesses and strengths of deep-learning contact restraints when coupled with advanced protein folding simulation algorithms.

## Results

**C-QUARK significantly outperforms QUARK in ab initio structure prediction.** Built on one of the top ab initio protein folding simulation programs, QUARK[5,30], C-QUARK starts with multiple sequence alignment (MSA) collection from whole-genome and metagenome sequence databases[32], where two types of contact-maps are created by deep-learning[29,33–36] and co-evolution[26,37–40] based predictors. Next, structural fragments with continuous sequence lengths (1-20 AA) are collected from unrelated PDB structures and used to assemble full-length structure models by Replica-Exchange Monte Carlo (REMC) simulations under the guidance of a composite force field consisting of knowledge-based energy terms, inter-residue contacts collected from the structure fragments based on their distance profiles[30], and the sequence-based contact-map predictions (Fig. 1).

Since the major difference between C-QUARK and QUARK is the incorporation of contact restraints in the former program, benchmarking the two programs can examine the effectiveness of contact-maps in ab initio folding of proteins. We collected a set of 247 non-redundant single domain proteins from the PDB, which had resolutions better than 3 Å and lengths between 50–300 residues (see Supplementary Data S2). Table 1 summarizes the folding results, where the average TM-score (template-modeling score) of the first models from C-QUARK (0.606) was 43% higher than that by QUARK (0.423). This difference corresponded to a p-value of $6.8 \times 10^{-51}$ as calculated by a one-sided Student's t-test, showing the improvement from the contact-map predictions is highly statistically significant. Table 1 also lists the results for the best in top-five models, which were ranked based on the cluster size of decoys from SPICKER[41], where C-QUARK once again significantly outperformed QUARK with an average TM-score (=0.629) that was 34% higher than the latter (=0.468) with a p-value of $2.0 \times 10^{-47}$. Here, TM-score is metric for assessing the structural similarity between the models and the native structures, and takes a value in the range (0, 1][42]. Statistics shows that a TM-score >0.5 corresponds to a model with a similar fold as the native[43].

To examine the advantage of C-QUARK on specific targets, we present a head-to-head TM-score comparison with QUARK in Fig. 2a, where the RMSD (Root-mean-square deviation) comparison of the two programs is listed in Fig. S1 in Supplementary Information (SI). The data show that C-QUARK generated better models with higher TM-scores (or lower RMSDs) than QUARK for 224 (212) out of the 247 targets. If we count the cases with correct structural folds, the first models by C-QUARK obtained correct folds for 186 (75% of the cases) targets, while only 71 targets (29% of the cases) were correctly folded by QUARK. The

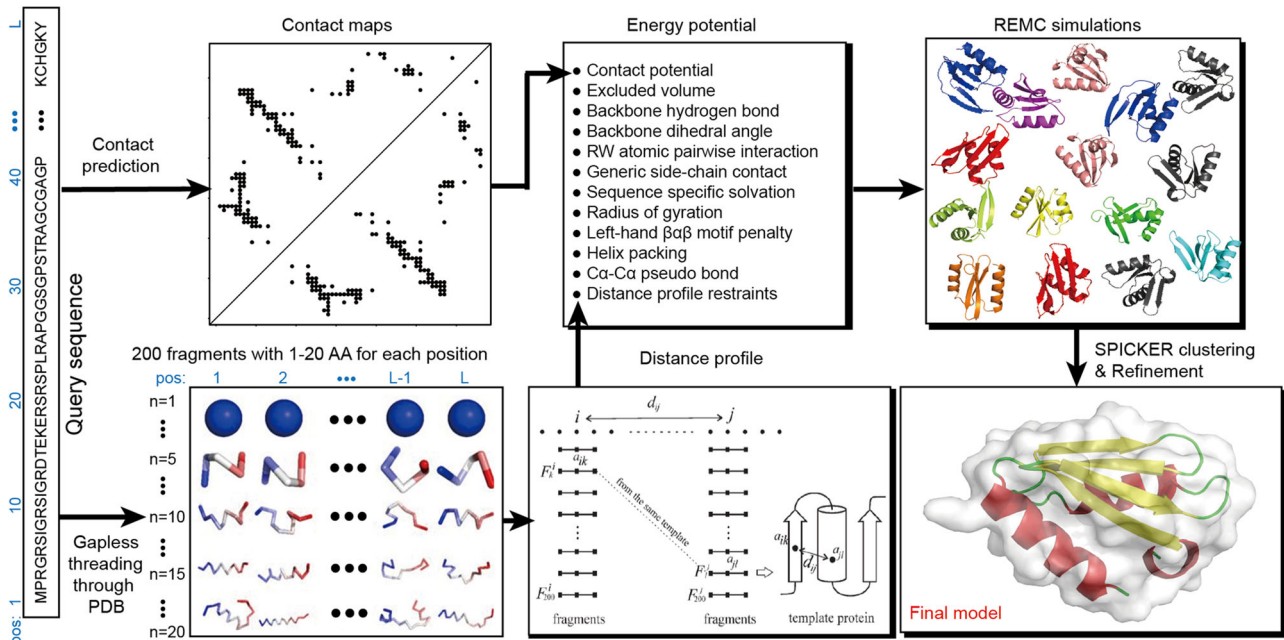

**Fig. 1 Flowchart of C-QUARK for contact-guided ab initio protein structure prediction.** The pipeline consists of five consecutive steps: multiple sequence alignment generation by DeepMSA, deep-learning based contact-map prediction, fragment creation, contact-guided Monte Carlo folding simulation, and model selection and refinement.

**Table 1 Summary of structure modeling by C-QUARK and QUARK on the 247 test proteins.**

| | First model | | Best in top-five models | |
|---|---|---|---|---|
| | **TM-score** | **RMSD (Å)** | **TM-score** | **RMSD (Å)** |
| C-QUARK | 0.606 (186, 75%) | 6.94 | 0.629 (196, 79%) | 6.22 |
| QUARK | 0.423 (71, 29%) | 12.14 | 0.468 (90, 36%) | 10.31 |
| *P*-value | $6.8 \times 10^{-51}$ | $1.8 \times 10^{-29}$ | $2.0 \times 10^{-47}$ | $9.1 \times 10^{-23}$ |

*P*-values are calculated between C-QUARK and QUARK using one-sided Student's *t*-tests. The values in parentheses represent the number and percentage of the cases with TM-scores >0.5.

number of correct folds generated by the two programs increased further to 196 and 90, respectively, when the best in top-five models was considered. It is noteworthy that there was no target for which QUARK generated a model with a TM-score ≥0.5 that C-QUARK did not do the same for. Rather, C-QUARK generated correct folds for 46% of the cases that were not foldable by QUARK, indicating the dominantly positive impact of contact restraints in ab initio folding of protein structures by C-QUARK.

To evaluate the ability of C-QUARK to model different protein types, we classified the test targets into three categories, *alpha*, *beta* and *alpha-beta* proteins, based on their secondary structure compositions (Supplementary Data S2). Although it is relatively easier to model *alpha*-proteins compared to other protein types as witnessed in previous CASP experiments, QUARK could generate correct folds for only 42% of the cases (24 out of the 64 *alpha* proteins, Table S1). On the other hand, the integration of contact restraints in the C-QUARK simulations resulted in correct folds for 52 out of the 64 (81%) *alpha* proteins, which was almost double that of QUARK. Furthermore, the success rates of C-QUARK for folding *beta* and *alpha-beta* proteins was ~3 fold higher than that of QUARK. For instance, out of the 67 *beta*-proteins and 116 *alpha-beta* proteins in the test set, C-QUARK

generated correct folds for 42 and 92 cases while QUARK did so for only 15 and 29 cases, respectively. The improvement of the modeling accuracy for *beta*-proteins is particularly exciting, since ab initio modeling of *beta*-proteins has been notoriously difficult as observed in CASP experiments[9–11]. The primary difficulty in folding *beta*-proteins lies in the fact that *beta*-proteins often have complicated topologies characterized by long-range contact-maps, where the inherent force fields of ab initio folding programs usually have difficulty capturing such long-range interactions formed by subtle hydrogen-bonding networks. The incorporation of long-range inter-residue contact prediction in C-QUARK effectively captured such interactions and significantly improved the folding performance for targets with complicated *beta*-fold topologies.

In Fig. 2b, we further examine the folding ability of C-QUARK and QUARK for proteins with different lengths. The average TM-scores of the C-QUARK models for proteins with lengths in the range 50–100, 101–50, 151–200, 201–250 and 251–300 residues were 0.588, 0.621, 0.638, 0.542 and 0.627, while those for QUARK were 0.516, 0.431, 0.388, 0.300 and 0.333, respectively. Obviously, the difference between the two programs increased as the size of the proteins increased, which is understandable as QUARK tends to have a relatively higher success rate for small proteins, while the contact-map accuracy and C-QUARK performance have no obvious length-dependence in this test set (Supplementary Fig. S2). To simplify the comparison, we also split the proteins into two sets of small proteins (with ≤150 residues) and large proteins (>150 residues). For the 156 small proteins, the average TM-scores of the C-QUARK and QUARK models were 0.607 and 0.467, corresponding to only a 30% TM-score improvement by C-QUARK. However, for the 91 large proteins, the improvement increased to 74%, where the average TM-scores were 0.604 and 0.347 for C-QUARK and QUARK, respectively. In addition, the average TM-score by C-QUARK was largely comparable for the small and large proteins (0.607 vs 0.604), while the modeling accuracy of QUARK was dramatically worse for large proteins (0.467 vs 0.347).

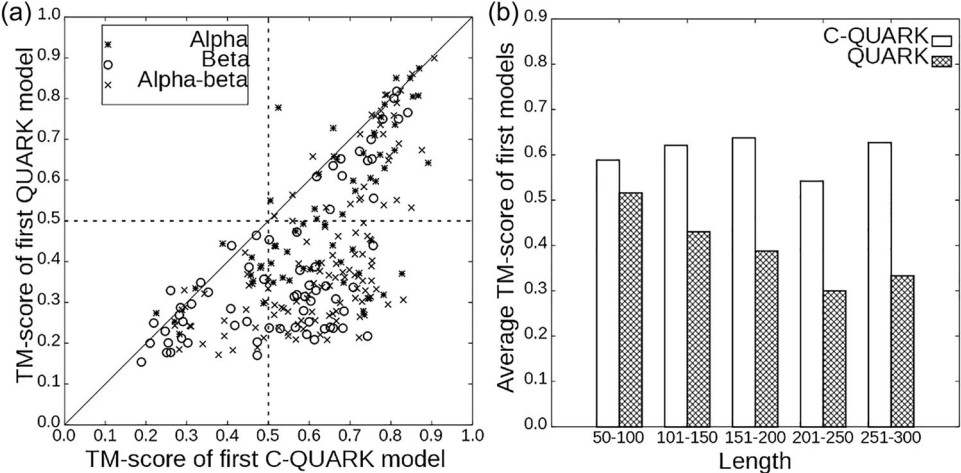

**Fig. 2 Comparison of C-QUARK and QUARK models on 247 test proteins. a** TM-scores of the first models by C-QUARK versus those by QUARK for different protein classes, where the dotted-crosses represent the alpha proteins, the circles represent the beta proteins, and the crosses indicate the alpha-beta proteins. **b** Average TM-scores by C-QUARK and QUARK at different protein-length intervals, where the white bars represent the C-QUARK results and the gray bars correspond to the QUARK results.

**Case studies reveal important roles of both medium- and long-range contacts on folding proteins with complicated topologies**. To investigate the reasons for the dramatic improvements, we present a structural comparison of the C-QUARK and QUARK models with the corresponding native structures along with the contact-map predictions for three test cases in Fig. 3. The first example (PDBID: 2d7jA) is a large *alpha-beta* protein with 188 residues that consists of 11 *beta*-strands, five alpha-helices and one $3_{10}$-helix. The core of the domain is a seven-stranded *beta*-sheet surrounded by *alpha*-helices on both sides (Fig. 3a). The native contact-map in Fig. 3d shows that the helices at the N-terminal ($H_N$) and the C-terminal ($H_C$) are in proximity due to long-range interactions between residues in the two helices (marked with rectangles in Fig. 3d). The majority of the native contacts, including the long-range contacts that hold the helices at both termini together, were correctly predicted, where the contact-map prediction accuracy was 0.648, as shown by the red circles in the left triangle of Fig. 3d. The restraints from these predicted contacts primarily lead to the arrangement of the residues in the C-QUARK model with the same contact network as the native structure, as shown in the C-QUARK model and in the contact-map with blue circles in the left triangle of Fig. 3d. The contact restraints in the core regions also helped maintain the overall topology of the seven *beta*-strands in that region. As a result, the C-QUARK model was very similar to the native with a TM-score=0.793. On the other hand, due to the lack of long-range contact restraints between the N- and C-termini in the QUARK simulations, the two *alpha* helices at the termini were far away from each other in the QUARK model. Hence, the overall fold and the corresponding contact-map (blue circles in the right triangle of Fig. 3d) of the QUARK model were significantly different from that of the native, resulting in a low TM-score of 0.295. This example highlights the importance of contacts, particularly long-range contacts, for correctly modeling large *alpha-beta* proteins.

Although *alpha*-proteins are relatively easier to model, we highlight in Fig. 3b how contacts can further improve modeling accuracies for *alpha*-proteins by not only bringing the contacting residues close to each other but also helping to reshape the helix structures and correct inter-helical orientations (parallel or anti-parallel). This target (PDBID: 1y9iA [https://doi.org/10.2210/pdb1Y9I/pdb]) has 159 residues and contains nine *alpha* helices, where Helix-9 at the C-terminal forms long-range contacts with

Helix-2, 3, 4 and 6. Additionally, Helix-9 is arranged in an anti-parallel fashion with Helix-3, 4 and 7, while it is parallel to Helix-2 and 6. Contacts predicted between Helix-9 and the five other helices, shown by the red circles in the left triangle of Fig. 3e, helped bring the five helices closer to Helix-9 during the C-QUARK simulations. Moreover, the correct prediction of these contacts contributed to the correct orientations of the helices as highlighted by the rectangles in the contact-map. Therefore, C-QUARK was capable of modeling the shape and orientation of all nine helices correctly and generated a model with a TM-score of 0.782. On the other hand, QUARK generated a wrong fold (TM-score=0.319) with incorrect packing and orientation of the helices due to the lack of contact information in the simulations.

While long-range contact predictions are especially helpful, it is noteworthy that medium-range contacts also play an important role in dictating ab initio protein folding. Figure 3c shows an example from the polysaccharide lyase-like protein (PDBID: 4peuA [https://doi.org/10.2210/pdb4PEU/pdb]) which has 250 residues and comprises 29 *beta*-strands. The first five strands from the N-terminal form two sandwiched anti-parallel *beta*-sheets, where Strand-1 forms long-range contacts with Strand-5, and short- and medium-range contacts with Strand-2. As a result, Strand-1 intervenes between Strands-2 and 5, where Strand-1 is anti-parallel to Strand-2 and parallel to Strand-5. Due to correctly predicted contacts between residues of Strand-2 and 5 and those of Strand-1, which are mainly in the medium-range as shown by the red circles and highlighted with rectangles (Fig. 3f), these N-terminal *beta*-sheets were all correctly formed during the C-QUARK simulations. The remainder of the C-terminal structure, which is comprised of eight helical turns of right-handed *beta*-helices, was also modeled correctly due to the restraints from the prevalent medium-range contacts (Fig. 3f). Overall, the C-QUARK model had a close structure to the native with a TM-score of 0.638, while the QUARK simulations only lead to short-range hydrogen-bonding and the formation of pairing between the adjacent *beta*-strands, which resulted in an incorrect fold with a TM-score of 0.236.

**C-QUARK performance correlates with contact prediction accuracy and satisfaction rate**. One of the critical factors for contact-assisted ab initio folding is the accuracy of predicted contacts used as restraints during the simulations. Figs. S3A and

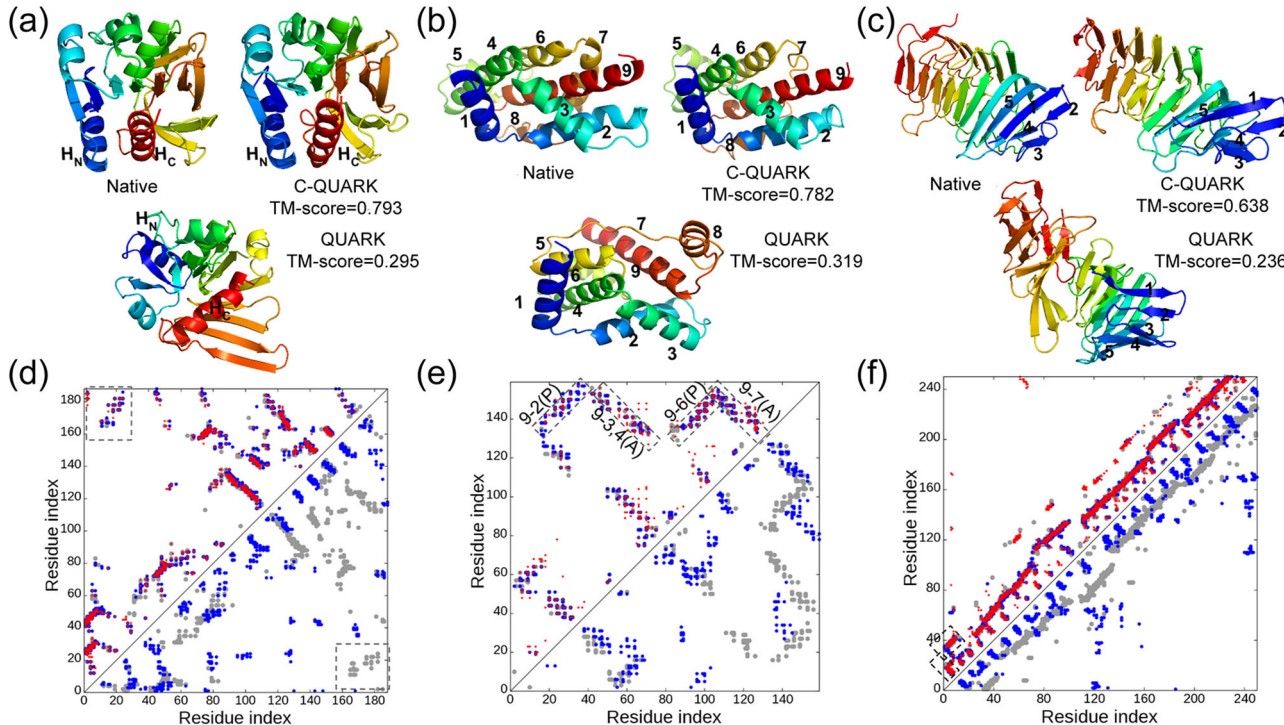

**Fig. 3 Illustrative examples for contact-guided ab initio structure folding. a–c** 3D structures for 2d7jA [https://doi.org/10.2210/pdb2D7J/pdb], 1y9iA [https://doi.org/10.2210/pdb1Y9I/pdb] and 4peuA [https://doi.org/10.2210/pdb4PEU/pdb], with blue to red running from N- to C-terminal.
**d–f** corresponding contact-maps for the native structures (gray circles), predicted contacts (red circles), the contacts in the C-QUARK models (blue circles in the upper left triangles), and contacts in the QUARK models (blue circles in the lower right triangle). In **a**, **d**, $H_N$ and $H_C$ represent the N- and C-terminal helices, respectively, which are in contact in the native structure and C-QUARK model but not in the QUARK model, as highlighted in rectangles in the contact-map. In **b**, **e**, numbers indicate the order of helices from N- to C-terminal, where the contacts that contribute to the correct helix packing are highlighted by rectangles with 'A' and 'P' referring to anti-parallel and parallel, respectively, in the contact-map. In **c**, **f**, numbers indicate the order of the beta-strands at the N-terminal whose contacts are highlighted by rectangles in the contact-map.

S3C show that the accuracy of the contact-maps extracted from the final C-QUARK models is indeed closely correlated with the accuracy of the predicted contact-maps that are used in the simulations, with a Pearson correlation coefficients (PCCs) of 0.703, 0.783, 0.778 and 0.793 for short-, medium-, long- and all-range contacts, respectively. These data imply that the contact restraints are effectively implemented in the folding simulations, which constitutes the major driving force for the success of C-QUARK ab initio structure modeling. The evidence can also be seen from the data in Supplementary Table S2, in which the accuracy of the overall input contacts to C-QUARK (0.502) was significantly higher than those extracted from the QUARK models (0.335) with a p-value of $1.2 \times 10^{-30}$. Interestingly, although the goal of C-QUARK is not to generate a high-accuracy contact-map, the contacts derived from the C-QUARK models had a slightly (but statistically significantly) higher accuracy than the input contacts, suggesting that the combination of deep-learning contacts with the QUARK force field can further improve the overall contact-map quality in the final models.

In Supplementary Fig. S3B, D, we present the accuracy of contact predictions directly against the TM-scores of the final C-QUARK models. While there is still a positive correlation between the TM-score and contact accuracy, it is considerably weaker than that of the contact-contact correlations (with PCC = 0.620 vs 0.793 for all-range contacts). In particular, among the 46 targets whose long-range contacts had low accuracy (<30%), C-QUARK created correct folds with a TM-score >0.5 for 20 cases (43%) (Supplementary Fig. S3D). In comparison, for these 46 targets, QUARK only successfully folded 9 proteins with TM-scores >0.5. These data demonstrate the ability of QUARK to

fold protein structures through its inherent fragment assembly procedure combined with fragment-based distance profiles[30]. Furthermore, the number of foldable cases increased by a factor of 2.2 even with low-accuracy contact-maps, showing the effectiveness of C-QUARK at incorporating noisy contact-maps in its folding simulations for ab initio structure prediction.

While accurate contact prediction helps improve the success rate of the ab initio folding simulations, it is also important to incorporate the contacts in the simulations in an efficient way. Since contact predictions with higher confidence scores generally have a higher likelihood of being correct (see Supplementary Fig. S4), our 3 G contact potential (Eq. 1 in Methods) was designed in a way that the well depth is proportional to the confidence score of each predicted contact (Eq. S4 in SI) and therefore the folding simulations primarily satisfy the contacts with higher confidence scores. Accordingly, a strong correlation was observed between the contact satisfaction rate and the accuracy of the predicted contacts with PCCs of 0.847, 0.798, 0.796 and 0.842 for the short-, medium-, long- and all-range contacts, respectively (Supplementary Fig. S5A). Such strong correlations in turn lead to a positive correlation between the contact satisfaction rate and the TM-scores of the final models, as shown in Supplementary Fig. S6, where the PCCs between the TM-scores of the final models and the contact satisfaction rates for all- and long-range contacts were 0.665 and 0.672, respectively.

As an illustration, Supplementary Fig. S7 lists the trajectories of the Replica-Exchange Monte Carlo (REMC) simulations for 1jiwI [https://doi.org/10.2210/pdb1JIW/pdb], where both the contact satisfaction rate and TM-score of the C-QUARK decoys increased

as the number of simulation cycles increased. As expected, the satisfaction rate of the predicted contacts during the QUARK simulation had no obvious change along with the simulation and the overall satisfaction rate was much lower than C-QUARK due to the absence of contact restraints in the simulation (Table S3). In Supplementary Fig. S8, we present the decoy conformations at the initial stages after the first REMC sweep and the final stage for target 1jiwI [https://doi.org/10.2210/pdb1JIW/pdb], where a correct fold with a TM-score=0.757 was produced by drawing the N- and C-terminal strands, which were initially 13.5 Å apart, to the native contact (~4.4 Å).

**C-QUARK significantly outperforms other contact-guided folding methods for targets lacking homologous sequences and high-accuracy contacts.** While contact-map prediction greatly improves the performance of ab initio folding, other physical and knowledge-based energy terms, including pairwise atomic potentials, solvation, hydrogen bonding, secondary structure element (SSE) packing and fragment-based distance profiles in C-QUARK (Eq. S2), also play important roles in improving modeling accuracy, e.g., by filtering out contacts that are physically unrealistic. Such complementarity between the contact potential and the inherent QUARK force field is vital in ab initio modeling. For instance, if the fragment-based distance-profile term is removed from the C-QUARK force field, the average TM-score of the first models by C-QUARK decreases from 0.606 to 0.593 with a p-value of $4.16 \times 10^{-4}$ (Table S4). Furthermore, if the entire fragments module, including the fragment-profile energy term and the fragment replacement movements in the simulation optimization (see details in Methods) is excluded from C-QUARK, the performance will become much worse, where the TM-score will be reduced from 0.606 to 0.553 with a p-value of $1.59 \times 10^{-30}$. These data indicate that the structural fragment module plays an important role in C-QUARK, which further demonstrate that the success of C-QUARK should be attributed to the interplay between predicted residue-residue contacts and the inherent force field and structural assembly simulation process.

To further quantitatively examine the importance of the comprehensive force field, we compared the performance of C-QUARK with three other programs that build structural models mainly based on predicted contacts or distances, including CNS[44], DConStruct[45] (v1.0) and trRosetta[7] (v1.0). Here, CNS constructs protein structures primarily based on the satisfaction of distance geometries. The DConStruct algorithm is similar to CNS, but also considers idealized secondary structure geometries and produces models using the Limited-memory Broyden–Fletcher–Goldfarb–Shanno[46] (L-BFGS) procedure found in the MODELLER (v9.21) package[47]. trRosetta builds the model with two steps. The first is L-BFGS energy minimization with a restrained version of Rosetta, where the restraints contain inter-residue distance and orientation distributions from deep residual neural network predictions. In the second step, statistical energy functions are added to the force field to relax the model. Here, we implement CNS through the CONFOLD (v1.0) package[48]. The input features for CNS and DConStruct are built on the same set of contact and secondary structure predictions as what are used in C-QUARK. Since trRosetta generates restraints on its own, we provided the same MSAs but used only the contact restraints (i.e., distances where the peak of the predicted distance distribution was lower than 8 Å or the sum of probabilities below 8 Å was greater than 0.5), to provide a fair comparison with C-QUARK.

The modeling results of C-QUARK, CNS and DConStruct on the 247 test proteins are summarized in Table S5, where the average TM-score of the first models by C-QUARK (0.606) was 14 and 16% higher than that of CNS (0.530) and DConStruct (0.524), respectively; the differences corresponded to p-values of $3.5 \times 10^{-20}$ and $1.5 \times 10^{-25}$ as determined by one-sided Student's t-tests. Figs. S9A and S9B present a head-to-head TM-score comparison between the methods, where the first models from C-QUARK had a higher TM-score than CNS (DConStruct) in 199 (198) out of the 247 cases, while the CNS (DConStruct) models did so for only 48 (49) of the cases. Notably, out of the 59 targets which had either a low effective number of sequences ($N_f < 15$) or a low contact-map accuracy (<30%), C-QUARK generated correct folds for 24 cases (i.e., 41% of the cases), while CNS (DConStruct) obtained correct folds for only 4 (4) of the cases (Table S6). Since contact prediction with low $N_f$ MSAs has been an bottleneck in contact-guided ab initio modeling[11], such a significantly increased success rate by C-QUARK in generating correct models for these challenging targets is particularly encouraging. Meanwhile, the TM-score of C-QUARK (0.428) for these 59 targets was also significantly (p-value $= 1.36 \times 10^{-6}$) higher than that of QUARK (0.348), showing that contact-map predictions are still helpful for folding despite the relatively lower accuracy (Supplementary Fig. S9C and Table S6).

Since the same contact-maps were used by all three programs, it is of interest to examine why C-QUARK could create models with obviously better quality, particularly for the cases with low $N_f$ and low contact prediction accuracy. Here, we used models produced by C-QUARK and CNS to highlight the reasons. Figure 4a and d show an example from 3teqB [https://doi.org/10.2210/pdb3TEQ/pdb], an *alpha*-protein packed with two anti-parallel, long helices. The $N_f$ value for this target was relatively low (=12.2), which resulted in the contact-map (red circles in Fig. 4d) being comprised of many falsely predicted contacts. Overall, the contact prediction accuracy was 0.273 and 0.213 for long- and all-range contacts, respectively. With the help of the SSE prediction and pair-wise atomic and helix packing interactions contained in the inherent C-QUARK force field, C-QUARK eliminated the majority of the false-positive contacts during the simulations, as observed in the contact-map of the final model in Fig. 4d (blue circles in the left triangle) with accuracies of 0.667 and 0.500 for long- and all-range contacts, respectively. As a result, C-QUARK generated a model with a similar fold to the native with a TM-score of 0.658, shown in blue in Fig. 4a. On the other hand, the helices in the CNS model (shown in green in Fig. 4a) were bent in an unrealistic fashion due to the satisfaction of false-positive contacts (blue circles in the right triangle of Fig. 4d), resulting in a model with a low TM-score (0.289). It is noted that without contact information, C-QUARK would not be able to obtain a correct model as the TM-score of the QUARK model was only 0.44 for this target, demonstrating again the importance of the complementarity of the QUARK force field and the contact restraints even at a low accuracy.

Figure 4b and e shows another example from 1zuuA, which is a small *beta*-protein with 56 residues. Here, the $N_f$ was very high (=1504.9), and hence the contact prediction accuracy for short-, medium, long- and all-range contacts was relatively high with accuracies of 0.6, 0.625, 0.659 and 0.627, respectively. The accuracies of the contact-maps derived from the final C-QUARK models increased further to 0.897, 0.836, 0.775 and 0.831, respectively, due to the removal of false positive contacts that clashed with the pairwise atomic interactions and hydrogen bonding between the *beta*-strands that formed the *beta*-sheets. As a result, the TM-score of the C-QUARK model for this target was 0.808. On the other hand, the TM-score of the CNS model was only 0.271, mainly due to false-positive contacts (highlighted by

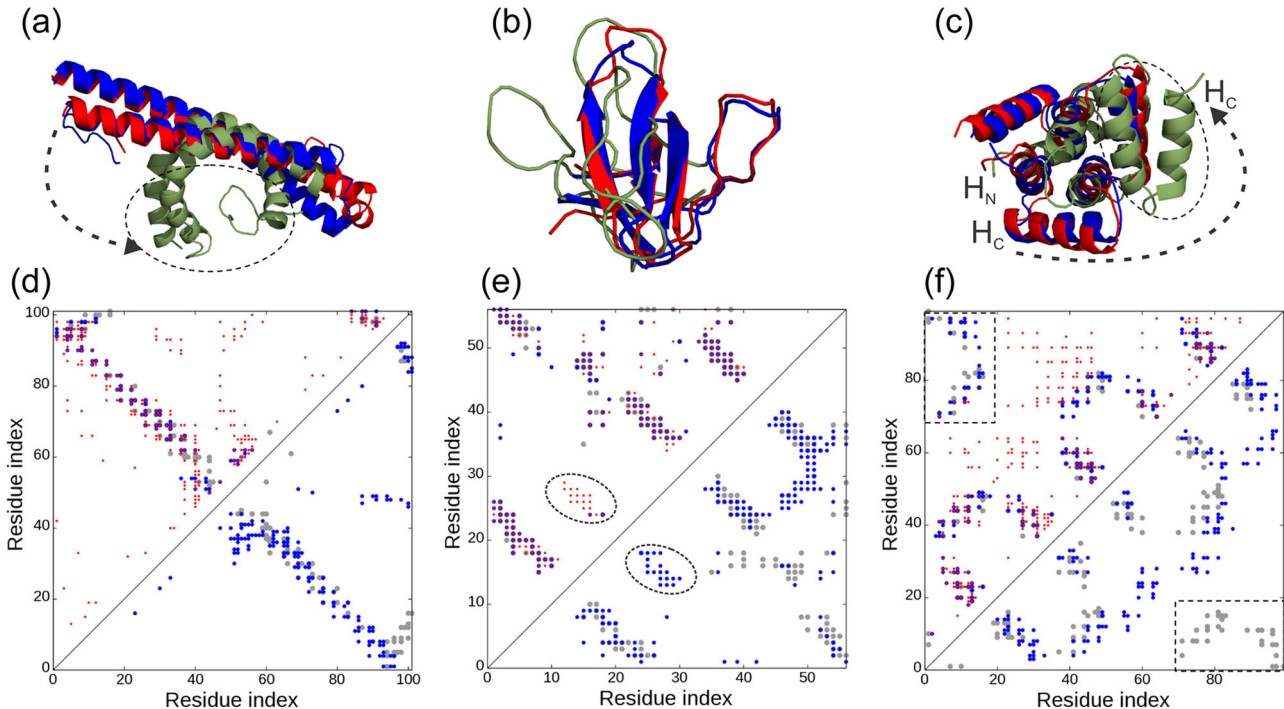

**Fig. 4 Illustrative examples for C-QUARK and CNS structure prediction on the same contact-maps. a–c** Overlay of predicted models (blue: C-QUARK; green: CNS) with the native (red) for 3teqB [https://doi.org/10.2210/pdb3TEQ/pdb], 1zuuA [https://doi.org/10.2210/pdb1ZUU/pdb] and 4yy2A [https://doi.org/10.2210/pdb4YY2/pdb], respectively. **d–e** Corresponding contact-maps for the native structure (gray circles), predicted contacts (red circles), contacts in the C-QUARK models (blue circles in the upper left triangles), and contacts in the CNS models (blue circles in the lower right triangle). In **a**, the dashed circle and arrow mark the unrealistic bending of the helices in the CNS model due to satisfying too many falsely predicted contacts. In **e**, the dashed circle highlights the falsely predicted beta-sheet that was filtered out by C-QUARK but not by CNS. In **c, f**, the dashed circle and arrow mark the incorrect move of the C-terminal helix (H_C) away from the N-terminal helix (H_C) due to absent contact predictions, while C-QUARK's inherent potential captured the inter-helix interactions as highlighted in the rectangular region of the contact-map.

the dashed circles in Fig. 4e) that were correctly filtered out by C-QUARK but that incorrectly guided the CNS modeling.

One of the hallmarks of C-QUARK is that even if contact restraints are not present for some region of the query, the inherent QUARK potential can often help compensate for their absence and create correct full-length models. Figure 4c shows such an example from 4yy2A [https://doi.org/10.2210/pdb4YY2/pdb], for which the native contacts between the N- and C-terminal helices (H_N and H_C) were not predicted (i.e., the red circles are largely absent in the rectangles in Fig. 4f). Additionally, due to the low $N_f$ (=0.402), numerous false positive contacts were scattered around the contact-map. Despite the lack of contacts in the helix regions and the use of noisy contact restraints, the inherent QUARK potential correctly captured the interaction of the terminal helices and generated a model with a correct fold and a high TM-score of 0.813. On the other hand, CNS generated a completely wrong model with a TM-score=0.290 by satisfying too many of the false positive contacts. In particular, due to the missing H_N-H_C contact restraints, the N-terminal helix was positioned far away from the C-terminal helix in the CNS model.

It is important to note that in the construction of our test dataset, homologous entries with sequence identities >30% to the training proteins of C-QUARK were filtered out. However, sequences homologous to the training sets of ResPRE and other third-party contact predictors, whose contact predictions are used by C-QUARK, were not particularly excluded from our test dataset. One reason is that the training sets for contact predictors are very large (e.g., the ResPRE training set included about 5,600 high-resolution protein structures and DeepContact utilized

around 14,000 proteins from SCOPe 2.06 to train the method, etc.,), to facilitate effective deep-learning training. Thus, the filtering of homologous proteins from these training sets would result in an insufficient number of proteins in the test dataset. Furthermore, C-QUARK, CNS and DConStruct utilized the same set of contacts, thus we did not specifically filter out the homologous proteins in the test set. However, since trRosetta generates spatial restraints using its own deep-learning predictor, to provide a fair comparison, we constructed a test dataset by removing proteins with a 50% sequence identity to not only the training sets of all the contact predictors used by C-QUARK, but also the training set of trRosetta. This resulted in only 57 proteins being left in this test dataset. Supplementary Table S7 shows the results for the modeling performance of C-QUARK, CNS, DConStruct and trRosetta on this reduced test set. The TM-score of the C-QUARK models on this reduced dataset was slightly lower than that of the entire test set (compared to Supplementary Table S6), probably due to the fact that this sub-dataset is non-redundant with the training set and is thus more difficult for contact prediction as the average accuracy was also reduced for CNS and DConStruct. Nevertheless, C-QUARK still significantly outperformed all the other control methods on this reduced dataset. It is notable that C-QUARK was 13.4% better than trRosetta, which was modified to only use predicted contacts derived from the distance predictions as restraints, in terms of the average TM-score of the first models (Supplementary Fig. S9D). Despite the fact that the relax/refinement step of trRosetta also uses physical and knowledge-based potentials, the global fold is primarily decided by the energy minimization step that only used predicted restraints. These results again demonstrate that

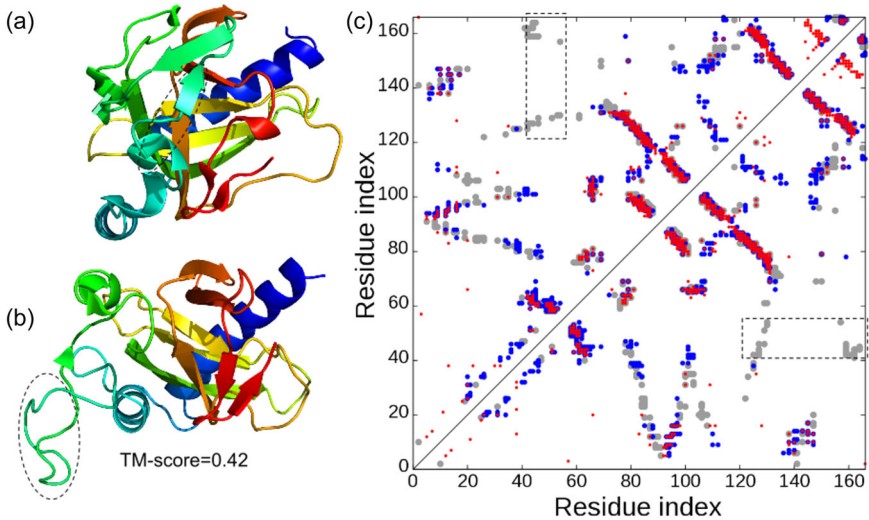

**Fig. 5 An example case (PDBID: 2xvsA [https://doi.org/10.2210/pdb2XVS/pdb]) for which C-QUARK failed to generate a correct fold. a** The native structure with a dashed circle highlighting the regions (40–43 and 51–52) which should be a $3_{10}$-helix and *beta*-strand, respectively. However, the secondary structure predictor, PSSpred, predicted these regions as coils. **b** The best in the top five models from C-QUARK with the dashed circle marking the coil region that was not folded correctly due to the wrong SSE prediction. **c** The contact-maps for the native structure (gray circles), C-QUARK model (blue) and sequence-based predictions (red), where the rectangles highlight the above-mentioned region which has no predicted contacts.

C-QUARK outperforms other contact-based folding programs, mainly due to the help from its comprehensive force field used in the structural assembly simulations.

**Why does C-QUARK fail for the remaining 13% of cases?** The examples in Figs. 3 and 4 have shown that the advanced force field can help in modeling the sequences where the contact restraints are incorrect or not available. This statement is mostly valid for the regions comprised of regular SSEs (*alpha*-helix or *beta*-sheet), as the QUARK force field has particular energy terms to enhance the packing of regular SSEs[5]. For cases that non- or incorrectly predicted contacts are involved in the loop or coil regions, the structure of which is much less regular, however, C-QUARK often fails to correctly model these regions. The data in Fig. 2a show that C-QUARK did not produce correct folds (TM-scores >0.5) for 51 out of the 247 test targets. Out of these 51 targets, 18 targets obtained reasonable folds with TM-scores ≥0.45, while models for the remaining 33 targets could be considered as incorrect. As shown in Supplementary Fig. S10, most of these targets had low contact accuracy, i.e., <0.4. Nevertheless, there were a few targets whose contact prediction accuracy for both long- and medium-range contacts was above 0.4. The reasons C-QUARK failed to fold these proteins is due to either mis-predicted SSEs or the lack of key contact predictions in the segments involving loops and coils.

Figure 5 is an example from 2xvsA [https://doi.org/10.2210/pdb2XVS/pdb], which shows that both these factors hindered the modeling accuracy of C-QUARK. This target is an *alpha-beta* protein that contains a $3_{10}$-helix and a *beta*-strand at the positions 40–43 and 51–52, respectively. However, the SSE prediction program, PSSPred[49], predicted the whole region [40–52] as a coil, as highlighted in Fig. 5a and b. Additionally, no contacts were predicted between residues in this and other regions, although long-range contacts exist and bring this region and the C-terminal residues in the native structure together as highlighted by the rectangle in Fig. 5c. Collectively, C-QUARK generated a poor model with a TM-score of 0.42 for this target. Similar issues also occurred for three other proteins, 1fasA [https://doi.org/10.2210/pdb1FAS/pdb], 3nikA [https://doi.org/10.2210/pdb3NIK/pdb] and 4h4nA [https://doi.org/10.2210/pdb4H4N/pdb], which had

medium- and long-range contact prediction accuracies >0.4 but TM-scores <0.45, as discussed in Supplementary Fig. S11. These examples highlight the importance of correct SSE prediction, where the development of specific potentials handling the loop and coil regions should be helpful to address this issue.

**Performance of C-QUARK on CASP13 targets**. Not all methods have standalone packages available. To compare C-QUARK directly with other state-of-the-art structure prediction programs, C-QUARK participated in the 13[th] critical assessment of structure prediction (CASP13) experiment as the "QUARK" server. Here, we analyzed the performance of C-QUARK on the 64 CASP13 FM (free modeling), FM/TBM (free modeling/template-based modeling) and TBM-hard (template-based modeling-hard) targets (Supplementary Data S3). By definition, these targets are challenging since homologous templates are absent or difficult to detect from the PDB library. Supplementary Table S8 lists the average TM-scores and GDT_TS (global distance test) scores of the first predicted models by C-QUARK and the other best five server groups in the CASP13 experiment. Here, GDT_TS is the standard score metric used by the CASP assessors. We collected the models directly from the "QUARK" server which ran C-QUARK.

This dataset should provide a relatively fairer test with state-of-the-art structure modeling programs since most programs used sequence-based contact-maps and all the programs had access to the most recent sequence databases released before CASP13. Based on the experimental structures of 64 CASP13 targets, the average GDT_TS of C-QUARK was higher than that of all other participating servers with p-values <0.05 as calculated by one-sided Student's *t*-tests (Supplementary Table S8). Especially in the TBM-hard and FM categories, C-QUARK was 4 and 5% better than the second-best method, respectively. For FM/TBM targets, BAKER-ROSETTASERVER (60.58) was slightly better than C-QUARK (58.94), but the difference was not statistical significantly. Some programs, such as the RaptorX servers, also used sequence-based distance map predictions[50]. It is notable that RaptorX-Contact predicted the residue-residue distances, and then fed the restraints into CNS to reconstruct the 3D models. The average GDT_TS score of the C-QUARK first models (52.09)

was still 12% higher than that of the RaptorX-Contact server (46.56). This gap was slightly smaller than the difference between C-QUARK and CNS in our benchmark test set (Supplementary Table S5, where C-QUARK was 16% better than CNS), which is probably because, compared to contact prediction, additional information can be extracted from distance predictions to help guide the CNS modeling in RaptorX-contact. In Supplementary Fig. S12, we highlight one of the FM targets, T0980s1-D1 (PDBID: 6gnxA [https://doi.org/10.2210/pdb6GNX/pdb]), which contains 105 residues with a 5-strand fold packed with an opposite helix. The TM-score of the first C-QUARK model was 0.540 for this domain, while the models generated by all other servers had TM-scores below 0.5. The poorer prediction for this target by other programs may be partially attributed to a low $N_f$ value (8.2) and, subsequently, low accuracy in predicted contacts with numerous false positives (highlighted by the rectangles in Supplementary Fig. S12B). On the other hand, most of the falsely predicted contacts were avoided in the C-QUARK models, due to the complementary effect of the inherent knowledge-based force field and the fragment-based distance profiles, which helped to correctly fold this target.

## Discussion

In this study, we developed a contact-guided ab initio folding program, C-QUARK, which showed a significantly improved ability to model "hard" proteins that do not have homologous templates in the PDB. While the C-QUARK pipeline is built on the platform of QUARK, one of the top ab initio modeling programs in the field, the average TM-score improved by 43% when the sequence-based contact predictions were incorporated in the pipeline. Importantly, the overall success rate for correct fold generation by C-QUARK was approximately 75%, which is 2.6 times higher than QUARK (29%), indicating the importance of contact-map prediction in improving ab initio structural modeling. Additionally, C-QUARK shows a consistent ability to fold medium- to large-sized proteins with lengths >150 residues which has been one of the limitations in the field of ab initio modeling for decades[51]. In the recent CASP13 experiment, for example, C-QUARK obtained models with TM-scores >0.50 for five ab initio modeling targets, T0950-D1 (PDBID: 6ek4A [https://doi.org/10.2210/pdb6EK4/pdb]), T0969-D1 (PDBID: 6cciA [https://doi.org/10.2210/pdb6CCI/pdb]), T0978-D1 [https://predictioncenter.org/casp13/target.cgi?id=72&view=regular], T1000-D2 (PDBID: 6u7lA [https://doi.org/10.2210/pdb6U7L/pdb]) and T1005-D1 (PDBID: 6q64A [https://doi.org/10.2210/pdb6Q64/pdb]), which had >300 residues.

The ability of C-QUARK to generate ab initio folds can be partly attributed to high-accuracy contact-maps by the deep-learning and coevolution-based predictors, as evident from the strong correlations (0.793 and 0.620) between the contact prediction accuracy and the contact accuracy and the TM-score of the final models, respectively. Since C-QUARK uses contact-maps from multiple programs built on different techniques[26,29,34–38], a key attribute of C-QUARK is the contact model selection procedure based on $N_f$, confidence score, residue separation and sequence length, which is implemented through a 3 G contact potential with parameters systematically optimized on the training dataset. The second important contribution to the success of C-QUARK is the effective fragment assembly simulations guided by the inherent knowledge- and physics-based force field extended from QUARK[5,30]. This contact-independent folding ability is critical to fold proteins that have low accuracy or unevenly distributed contact-maps, since the advanced folding simulations help to handle the regions without contact-map restraints. If we removed the fragment module during the folding simulations, the performance of C-QUARK would be reduced by approximately

10%. Moreover, the complementary energy terms in the inherent force field were found to be helpful for filtering out some of the false-positive contacts, including contacts that were not physically realistic or even those that were physical realistic but were inconsistent with the force field used during the folding simulations. This advantage was demonstrated on the benchmark data with the contact geometry-based structure construction programs, including CNS and DConStruct, where C-QUARK folded 6 times more targets with TM-scores >0.5 for the 59 targets with either shallow MSAs ($N_f < 15$) or low contact accuracy (<30%).

Nevertheless, there are several aspects of C-QUARK that may be improved. First, although the inherent QUARK force field can assemble the regular SSE regions for the cases when the contact accuracy is low, the modeling ability is much less efficient when the low accuracy contact regions involve loops and coils, which can be exaggerated when the SSEs are mis-predicted. Correct secondary structure prediction and development of specific energy terms for the loop/turn/coil residue packing will be important for addressing this issue. Second, the dynamics of the C-QUARK simulations are still unsatisfactory and often hinder the modeling accuracy of large proteins in a limited time, although the folding results do not obviously depend on the protein length when a sufficient number of simulation cycles are conducted (typically using 500 REMC sweeps)[5]. As shown in Supplementary Fig. S13, when the simulations are terminated after 50 h as set in the online server, for instance, the default simulations with 500 REMC sweeps cannot be completed for proteins larger than 230 residues; this is mainly due to the inefficient energy calculation process that takes approximately 55% of the simulation time. Further improvement of the energy calculation and optimization of the Monte Carlo (MC) movements can help speed up the simulation processes. Third, although C-QUARK outperformed most of the servers, including those using distance restraints, in CASP13, the most recent progress of the field showed advancement in modeling accuracy using deep-learning distance, inter-residue torsion angle and hydrogen bonding restraints for ab initio structure predictions[6,7,31,52]. Due to the limited information provided by the binary distance classification in contact prediction, folding programs that solely use contact restraints may not be comparable with the most advanced programs that combine contact restraints with those categories of spatial restraints (see the comparisons between C-QUARK, AlphaFold and trRosetta in Supplementary Table S9 on the 64 CASP13 FM targets); most of these components are yet to be contained in the current C-QUARK pipeline. Finally, modeling multi-domain proteins is much harder than folding single-domain structures because of the introduction of additional degree of freedom in inter-domain orientations. For instance, the average TM-score (0.47) of the full-length models predicted by C-QUARK for the 21 multi-domain targets in CASP13 was much lower than that (0.65) for the individual domains (Supplementary Fig. S14 and Table S10). This is mainly due to the low accuracy of inter-domain contact prediction compared to intra-domain contact prediction, where the low contact accuracy probably originates from the worse MSA quality for the multi-domain sequences. Meanwhile, many energy terms of the C-QUARK force field, including solvation and radius of gyration, have been designed and optimized for single-domain structure folding. Overall, while there is considerable space for further improvement and many of the strategies/components are under development in C-QUARK, the results reported in this study demonstrate a robust and significant advantage of efficiently combining contact-map restraints with the cutting-edge folding assembly simulations for folding non- and distantly-homologous proteins.

## Methods

The C-QUARK pipeline is established on the framework of the QUARK ab initio structure prediction pipeline[5], where the flowchart is depicted in Fig. 1. Compared to the QUARK pipeline, C-QUARK has 3 major implementations, including: (i) a multiple sequence alignment generation tool, DeepMSA[32], which is used in C-QUARK for profile construction and contact-map prediction; (ii) the deep-learning-based and coevolution-based contact prediction module for residue-residue contact-map prediction, combination and selection; (iii) a contact potential term developed and carefully trained to balance its contribution with the other energy terms, including the inherent knowledge and physics-based potentials, in order to guide the structure assembly simulations. The primary components in the C-QUARK pipeline are described below.

**Multi-program contact prediction**. Starting from a query sequence, MSAs are generated by DeepMSA[32], which performs sequential searches through two whole-genome sequence databases (UniClust30 and UniRef90) and a metagenome sequence database (Metaclust)[53]. Next, contact predictions are generated using ten state-of-the-art contact predictors: NeBcon[37] (v1.0), ResPRE[29] (v1.0), DeepPLM[33] (v1.0), DeepCov[34] (v1.0), Deepcontact[35] (v1.0), DNCON2[36] (v1.0), MetaPSICOV2[38] (v1.0), GREMLIN[26] (v2.01), CCMpred[39] (v1.0) and FreeContact[40] (v1.0.21). We found that instead of using contacts from a single predictor, it is advantageous to use multiple predictors to improve the accuracy of the overall contact-maps. For instance, while the accuracies for the top $L$ and $L/2$ long-range predicted contacts by the best predictor, ResPRE, as shown in Supplementary Tables S11 and S12, are 0.538 and 0.685, respectively, these accuracies increase to 0.561 and 0.692, respectively, when contacts are ranked based on the consensus from multiple predictors. As a result, the modeling accuracy also increases when contacts from multiple predictors are used as restraints, as shown in Supplementary Fig. S15.

**Contact-map combination and selection**. The selection of correct contacts from different programs is essential for C-QUARK. To train the procedure, we collected a non-redundant set of 243 training proteins that had sequence identities below 30% to the 247 test proteins in this study (see Supplementary Data S1). Based on the average performance of the training proteins (Supplementary Tables S13 and S14), we classified the contact predictors into four categories: (i) NeBcon, ResPRE and DeepPLM as "very high", (ii) DeepCov, Deepcontact and DNCON2 as "high", (iii) MetaPSICOV2 as "medium", and (iv) GREMLIN, CCMpred and FreeContact as "low". Accordingly, C-QUARK selects more contacts from the predictors with higher accuracies. In addition, C-QUARK requires that any of the selected contacts must have a confidence score higher than a certain cutoff, which corresponds to an average accuracy of 50% in the training dataset. This cutoff is predictor-specific and depends on the effective number of sequences ($N_f$) in the MSAs[32] and the length of the query sequence (see Texts S1 and S2 in SI). On average, around $2.4*L$ contacts are selected for each target and used as restraints in the fragment assembly simulations.

**Fragment generation and REMC simulations**. Similar to QUARK[5,30], the query sequence is scanned through a non-redundant set of 29,156 high-resolution PDB structures by gapless threading to create a set of position-specific fragment structures with continuous lengths ranging from 1 to 20 residues. A histogram of distances $d_{ij}$ for each residue pair ($i$ and $j$) of the query is derived from the top 200 fragments at the $i$th and $j$th positions if the fragments are from the same PDB structure. The histogram that has a peak at the position of $d_{ij}<9$ Å is converted to a distance profile for the residue pair. The distance profile and contact-map restraints are combined with the inherent knowledge-based and physical energy terms and used to guide the fragment assembly simulations to construct full-length models. We note that to exclude potential contamination from homologous proteins, we removed all protein structures that had >30% sequence identities to the query from the template library during the generation of position-specific fragments.

For each target, five REMC simulations starting from different random numbers are run in C-QUARK. Forty replicas are implemented in each simulation, and the conformations in adjacent replicas are swapped following the Metropolis criterion after a cycle comprised of ($30*L^{1/2}$) MC movements. The MC movement sets consists of 11 local movements that can further be divided into three levels: residue level (M1–M4), segmental level (M5–M8), and topology level (M9–M11) as shown in Supplementary Fig. S16. Five hundred cycles are performed in the simulations by default. However, the simulations are forcefully terminated after 50 CPU hours even if the assigned cycles are not completed to ensure models are generated within a reasonable amount of time. Next, "Decoy" conformations from the simulation trajectories are clustered by SPICKER[41] to identify the largest clusters, which correspond to the lowest free-energy states. The cluster centroids are further refined by fragment-guided molecular dynamics (FG-MD)[54] to obtain the final models.

**C-QUARK force field**. The C-QUARK force field contains twelve energy terms as described in Eq. S2 in Supplementary Text S3. While most of the energy terms were extended from the QUARK force field with appropriate re-parameterization, the major term accounts for the predicted contact-map restraints and is defined with a

3-gradient (3 G) form (Supplementary Fig. S17):

$$E_{con}\left(d_{ij}\right) = \begin{cases} -U_{ij}, & d_{ij} < 8\text{ Å} \\ -\frac{1}{2}U_{ij}\left[1 - \sin\left(\frac{d_{ij}-(\frac{8+D}{2})}{d_b}\pi\right)\right], & 8 \le d_{ij} < D \\ \frac{1}{2}U_{ij}\left[1 + \sin\left(\frac{d_{ij}-(\frac{D+80}{2})}{(80-D)}\pi\right)\right], & D \le d_{ij} \le 80\text{ Å} \\ U_{ij}, & d_{ij} > 80\text{ Å} \end{cases} \quad (1)$$

where $d_{ij}$ is the $C_\beta$-distance between the residue pair ($i$, $j$). The depth of the potential, $U_{ij}$, is proportional to the confidence score of the contact prediction and calculated by Eq. S4 in Supplementary Text S4.

Overall, the 3 G potential contains a negative well at an 8 Å cutoff, with a strong force from 8 Å to $D$ ($=8 + d_b$), followed by a weaker force from $D$ to 80 Å being introduced to push the target residue pairs towards the well when they are a long distance apart (Supplementary Fig. S17). Here, the gradient width ($d_b$) of the contact well is the only free parameter of the 3 G potential which depends on the protein size and determines the convergence speed and satisfaction rate of the contact-maps in combination with the inherent QUARK potential. As shown in Supplementary Table S15, $d_b$ is typically narrow, e.g., 6 Å, when the length of the target is relatively small, e.g., < 100. On the other hand, the gradient width increases to 12 Å when the length is >200, since simulations with larger proteins are more difficult to converge and C-QUARK needs to use a wider well to draw the candidate residue pairs that are further apart in distance to the well smoothly and bring the residues pairs within 8 Å quickly. It is important that Eq. (1) is designed in a way that the potential curve is continuous and smooth (with $\partial E/\partial d = 0$) at all three transition points of $d_{ij} = 8$, $D$ and 80Å, so that the contact restraints can be implemented smoothly without singularities. Furthermore, since contact prediction can only tell whether the distance between a residue pair $i$-$j$ is below 8 Å or not, we designed the 3 G potential as a constant when the distance is < 8 Å. As almost all of the residue-residue distances in a normal size protein are lower than 80 Å, the potential is also designed as flat beyond the maximal distance threshold (80 Å). However, between 8 Å and 80 Å, we set the potential as two regions split at the transition point ($d_{ij} = D$). In the region above D, a relatively weaker force is used to avoid structural overpacking due to false positive contact predictions, while in the region below D, a stronger force is used to quickly satisfy the contact restraints since in this region the contact accuracy of the target residue pairs should be higher than that in the longer-distance regions (-because most of the adjacent residue pairs in the structure decoys should be more consistent with the inherent QUARK potential after the equilibrium obtained by the Monte Carlo simulations). Two trigonometric function style potentials are selected in the two regions to connect the flat areas, since trigonometric functions are simple, continuous, smooth, and differentiable.

Besides the developed contact energy term (3 G potential), the other energy terms have also been adjusted to maximize the folding performance of the 243 training proteins. For instance, the weight ($w_7$) of the distance-profile energy term ($E_{dp}$ in Eq. S2) was increased from 0.60 to 3.00 in the C-QUARK force field to allow the fragment-based potential to help filter out false positive contacts. Furthermore, we added an energy term, which accounts for the distance between adjacent Cα atoms ($E_{c\alpha}$ in Eq. S2 and Eq. S3), to penalize adjacent residue pair with Cα-Cα distances > 4 Å. This term is specifically designed to penalize backbone breaks that can occur after fragment movements, as a stronger trend of bond-breaking was seen after the introduction of contact predictions in C-QUARK.

**Statistics and reproducibility**. All experiments can be reproduced by running the different software. The C-QUARK results can be reproduced by using our server https://zhanggroup.org/C-QUARK/ or similar results can be obtained using the standalone package https://github.com/jlspzw/C-QUARK. The statistical data analysis is produced by R (4.0.3).

**Reporting summary**. Further information on research design is available in the Nature Research Reporting Summary linked to this article.

## Data availability

All datasets used in this study, including training, testing and CASP13 proteins are available in the PDB database (https://www.rcsb.org) and CASP official website (https://www.predictioncenter.org) under the accession codes provided in Supplementary Data S1–3 accompanying this manuscript. The datasets can also be downloaded from https://zhanggroup.org/C-QUARK/ or https://github.com/jlspzw/C-QUARK.

## Code availiability

The online server and standalone package of C-QUARK are made freely available at https://zhanggroup.org/C-QUARK/ and GitHub[55] (https://github.com/jlspzw/C-QUARK).

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

## Acknowledgements

We thank Dr. Baoji He for the initial preparation of the project. This work is supported in part by the National Institute of General Medical Sciences (GM136422, OD026825), the National Institute of Allergy and Infectious Diseases (AI134678), and the National Science Foundation (IIS1901191, DBI2030790, MTM2025426). This work used the Extreme Science and Engineering Discovery Environment (XSEDE), which is supported by National Science Foundation (ACI1548562).

## Author contributions

Y.Z. conceived and designed research; S.M. and Y.Z. wrote algorithm; S.M. and W.Z. performed experiment; S.M. and W.Z. analyzed data; C.Z. and W.Z. developed MSA

programs, Y.L. developed contact prediction programs; C.Z. constructed webserver; S.M., W.Z., C.Z., R.P. and Y.Z. wrote manuscript.

## Consent for publication

All authors have approved the manuscript for submission.

## Competing interests

The authors declare no competing interests.
