## [Peer Review File · Nature Communications]

Reviewers' Comments:

Reviewer #1:

Remarks to the Author:

C-Quark is a nice and new implementation for the improvement of ab-initio folding predictions. A relevant point is the efficacy of the new energy function which apparently allows the correct prediction of the structure of proteins starting from their sequence with very few homologous and with sparse predicted contact maps.

Major observations:

It is not perfectly clear to which extent the new energy function differs from the previous ones already present in QUARK. It is evident that the new implementation overpasses the previous one. However some more details will add to the validity of the method

Is there any other possible method to compare with? Starting from the observation that Quark is already top category, according to CASP benchmarks, C-Quark is only scored against Quark in present paper. Also this should be discussed.

Reviewer #2:

Remarks to the Author:

The manuscript by S. M. Mortuza et al. describes a new folding method C-QUARK for protein structure prediction. The method is a significant extension of QUARK, an excellent software in the field. C-QUARK extends the QUARK force field with predicted contact map and then uses the extended force field to guide Monte Carlo fragment assembly simulations. One of the key elements of C-QUARK is 3G contact potential that selects contacts predicted by multiple programs. Experimental results suggest that C-QUARK outperforms QUARK and CNS in ab initio protein structure prediction, especially for hard targets, i.e., the beta/alpha-beta proteins with complicated topologies or low Nf value. Another advantage of C-QUARK is its robustness to the falsely predicted contacts (and even corrects the false-positive contacts), which is mainly due to the complementation between contacts and the QUARK force field. I also appreciate the failure analysis of C-QUARK, which is very interesting and instructive for further improvement.

Major comments:

1. The comparison is not enough to show the advantage of C-QUARK over other contact/distance-based methods. The authors shall compare their method with advanced contact-based approaches, such as AlphaFold and trRosetta.
2. The construction of the benchmark dataset (lines 95-97) is not very clear. Are these 247 test proteins collected in a fair way? Are they already in the training set of the used contact predictors (e.g. ResPRE and DeepCov)? The authors shall clearly indicate the potential overlap between test and training sets for all used contact predictors.
3. The performance for free-modeling targets is ambiguous. The authors shall evaluate CASP13 FM, FM/TBM, and TBM-hard targets separately.
4. The comparisons over CASP10-12 targets are not very fair. The authors use newer sequence and structure databases, thus obtain significantly better contacts than other predictors. It is recommended to remove these comparisons.
By the way, are these CASP10-12 targets excluded from the training set for all used contact predictors?
5. It is interesting to know how fragments affect performance as fragments key parts of C-QUARK. Although fragment-assembly has been shown successfully in QUARK, it is expected to show the necessity of fragments when high-quality contacts are given. I recommend the authors to carry out the following experiments:
 - a. To show the effects of fragments in the score function, the author might compare C-QUARK with a baseline model that discards all fragment-related energy items (e.g. fragment-based distance profile)

in the C-QUARK force field.

b. To show the effects of fragments in the optimization method, the author might compare C-QUARK with a fragment-free optimization method, such as gradient descent (as the contact-map energy function is differentiable).

6. It is hard to judge the advantage of C-QUARK when lacking homologous sequences. For the 59 targets with low accuracy contacts (line 254-263), the authors compared C-QUARK with CNS. However, even if the authors have provided case studies, it is still not clear whether C-QUARK could perform better than QUARK with low-quality contacts. The authors shall explicitly show the performance of QUARK over these 59 targets.

7. The performance of CNS is inconsistent with that of RaptorX, which uses CNS to construct prediction models. The authors showed C-QUARK performs better than CNS (0.606 vs 0.530 in TM-score); however, C-QUARK didn't show such a superiority over RaptorX (51.396 vs 49.457 in GDT_TS). An in-depth examination of this issue is expected.

8. The 3G contact potential term (Eq. 1) is the key element of C-QUARK; however, this term seems ad hoc. What are the intuition and physics underlying this term?

9. The authors stated that besides 3G contact potential, another key contribution to the success of C-QUARK is the effective fragment assembly simulations. Is there any significant difference between the fragment assembly strategies used by QUARK and C-QUARK?

Minor comments:

1. Why only single-domain proteins are chosen? Can C-QUARK build structures for multi-domain proteins? Further experiments on these proteins will be very instructive.

Response to Reviewer #1

We very much appreciate the comments and suggestions from the Reviewer, which we found very helpful for improving the quality of the manuscript. The major concerns from the Reviewer were regarding details of the C-QUARK pipeline and comparison of C-QUARK with more methods. In the revision, we have added significantly more details in the methods section and selected more contact/distance-based folding methods to compare C-QUARK to. In the following, we include point-by-point replies to the comments of the Reviewer, where all changes have been highlighted in yellow in the manuscript.

1. The Reviewer commented:

C-Quark is a nice and new implementation for the improvement of ab-initio folding predictions. A relevant point is the efficacy of the new energy function which apparently allows the correct prediction of the structure of proteins starting from their sequence with very few homologous and with sparse predicted contact maps.

We appreciate the positive comments from the Reviewer on the work.

2. The Reviewer commented:

Major observations:

It is not perfectly clear to which extent the new energy function differs from the previous ones already present in QUARK. It is evident that the new implementation overpasses the previous one. However some more details will add to the validity of the method.

Thank you for the question and suggestion. The C-QUARK force field contains twelve energy terms as described in **Eq. S2** in **Text S3**. While the first ten energy terms in Eq. S2 are extended from the QUARK force field, the major new term accounts for deep-learning contact-map restraints and is defined by 3-gradient (3G) potential. Besides this 3G potential, we also added a new energy term, which accounts for the distances between adjacent C α atoms ($E_{C\alpha}$ in **Eq. S2** and **Eq. S3**), in order to penalize chain breaking, i.e., with adjacent residue pairs with a C α -C α distance $> 4\text{\AA}$, as we found that this happens more often after the introduction of contact-maps. Finally, all the parameters and weight factors have been re-balanced and optimized after the introduction of the new contact maps.

To clarify the differences in the energy terms, we have partly rewritten the “**C-QUARK force field**” section in page 11-12 of the **Main Text**, and **Text S3** in the **SI** entitled “**Text S3. C-QUARK force field used to guide the REMC simulations**” as following:

Main Text:

C-QUARK force field. The C-QUARK force field contains twelve energy terms as described in **Eq. S2** in **Text S3**. While most of the energy terms were extended from the QUARK force field with appropriate re-parameterization, the major new term accounts for the predicted contact-map restraints and is defined with a 3-gradient (3G) form (**Fig. S17**):

$$E_{con}(d_{ij}) = \begin{cases} -U_{ij}, & d_{ij} < 8\text{\AA} \\ -\frac{1}{2}U_{ij} \left[1 - \sin\left(\frac{d_{ij} - \left(\frac{8+D}{2}\right)\pi}{d_b}\right) \right], & 8\text{\AA} \leq d_{ij} < D \\ \frac{1}{2}U_{ij} \left[1 + \sin\left(\frac{d_{ij} - \left(\frac{D+80}{2}\right)\pi}{(80-D)}\right) \right], & D \leq d_{ij} \leq 80\text{\AA} \\ U_{ij}, & d_{ij} > 80\text{\AA} \end{cases} \quad (1)$$

where d_{ij} is the C_β -distance between the residue pair (i, j). The depth of the potential, U_{ij} , is proportional to the confidence score of the contact prediction and calculated by **Eq. S4** in **Text S4**.

Overall, the 3G potential contains a negative well at an 8 Å cutoff, with a strong force from 8 Å to $D (=8\text{\AA} + d_b)$, followed by a weaker force from D to 80 Å being introduced to push the target residue pairs towards the well when they are a long distance apart (**Fig. S17**). Here, the gradient width (d_b) of the contact well is the only free parameter of the 3G potential which depends on the protein size and determines the convergence speed and satisfaction rate of the contact-maps in combination with the inherent QUARK potential. As shown in **Table S15**, d_b is typically narrow, e.g., 6 Å, when the length of the target is relatively small, e.g., < 100. On the other hand, the gradient width increases to 12 Å when the length is >200, since simulations with larger size proteins are more difficult to converge and C-QUARK needs to use a wider well to draw the candidate residue pairs that are further apart in distance to the well smoothly and bring the residues pairs within 8 Å quickly. It is important that **Eq. 1** is designed in a way that the potential curve is continuous and smooth (with $\partial E/\partial d = 0$) at all three transition points of $d_{ij} = 8, D$ and 80 Å, so that the contact restraints can be implemented smoothly without singularities. Furthermore, since contact prediction can only tell whether the distance between a residue pair $i-j$ is below 8 Å or not, we designed the 3G potential as a constant when the distance is < 8 Å. As almost all of the residue-residue distances in a normal size protein are lower than 80 Å, the potential is also designed as flat beyond the maximal distance threshold (80 Å). However, between 8 Å and 80 Å, we set the potential as two regions split at the transition point ($d_{ij} = D$). In the region above D , a relatively weaker force is used to avoid structural overpacking due to false positive contact predictions, while in the region below D , a stronger force is used to push contact restraints quickly satisfied since in this region the contact accuracy of the target residue pairs is supposed to be higher than that in the longer-distance regions (-because most of the adjacent residue pairs in the structure decoys are supposed to be more consistent with the inherent QUARK potential after the equilibrium of Monte Carlo simulations). Two trigonometric function style potentials are selected in the two regions to connect the flat areas, since trigonometric functions are simple, continuous, smooth, and differentiable.

Besides the newly developed contact energy term (3G potential), the other energy terms have also been adjusted to maximize the folding performance of the 243 training proteins. For instance, the weight (w_7) of the distance-profile energy term (E_{dp} in **Eq. S2**) was increased from 0.60 to 3.00 in the C-QUARK force field to allow the fragment-based potential to help filter out false positive contacts. Furthermore, we added a new energy term, which accounts for the distance between adjacent $C\alpha$ atoms ($E_{c\alpha}$ in **Eq. S2** and **Eq. S3**), to penalize adjacent residue pair with $C\alpha$ - $C\alpha$ distances > 4Å. This term is specifically designed to penalize backbone breaks that can occur after fragment movements, as a stronger trend of bond-breaking was seen after the introduction of contact predictions in C-QUARK.

Text S3:

Text S3. C-QUARK force field used to guide the REMC simulations

In order to guide its REMC simulations, C-QUARK uses the following force field that calculates the total energy of a conformation by summing up 12 energy terms¹²:

$$E_{tot} = w_1 E_{prm} + w_2 E_{prs} + w_3 E_{ev} + w_4 E_{hb} + w_5 E_{sa} + w_6 E_{dh} + w_7 E_{dp} + w_8 E_{rg} + w_9 E_{bab} + w_{10} E_{hp} + w_{11} E_{c\alpha} + w_{12} E_{con} \quad (S2)$$

Here, the terms account for the backbone atomic pairwise potential (E_{prm}), side-chain center pairwise potential (E_{prs}), excluded volume (E_{ev}), hydrogen bonding (E_{hb}), solvent accessibility (E_{sa}), backbone torsion angles (E_{dh}), fragment-based distance profiles (E_{dp}), radius of gyration (E_{rg}), strand-helix-strand packing (E_{bab}), helix packing (E_{hp}), distance between adjacent C α atoms ($E_{c\alpha}$), and the contact potential (E_{con}). While the first ten terms are used in both QUARK and C-QUARK, the final term, E_{con} , is unique to the C-QUARK force field and accounts for the contact restraints from the predicted contacts (see **Eq. 1** and **Fig. S17**). In addition to the contact potential term, the 11th energy term, which factors in the distance between adjacent C α atoms ($E_{c\alpha}$), is also a newly added term and takes the following form:

$$E_{c\alpha} = \sum_{i=1}^{L-1} I[d_{i,i+1} > 4](d_{i,i+1} - 4)^2 \quad (S3)$$

where $d_{i,i+1}$ is the C α -C α distance between residues i and $i+1$, and $I[]$ is the Iverson bracket, i.e., $I[d_{i,i+1} > 4] = 1$ if $d_{i,i+1} > 4$, and 0 otherwise. This term is designed to penalize backbone breaking with adjacent residue pairs with C α -C α distances $> 4\text{\AA}$ which can occur after fragment movements. All the weighting parameters in C-QUARK were re-tuned on the training protein set listed in **Dataset S1**, to appropriately balance the inherent force field with the contact restraints by maximizing the TM-score of the predicted models. As a result, most of the weighting parameters in w_{1-10} are similar to what was used in QUARK¹² despite the use of different training proteins, showing the robustness of the QUARK force field. It is interesting that the weight (w_7) of the distance-profile energy term increased from 0.60 to 3.00 in the C-QUARK force field to enlarge the effect of filtering out false positive contacts. The last parameter w_{12} is equal to 0.426 when $N_f > 50$, and 0.355 otherwise.

3. The Reviewer commented:

Is there any other possible method to compare with? Starting from the observation that Quark is already top category, according to CASP benchmarks, C-Quark is only scored against Quark in present paper. Also this should be discussed.

Thank you for the suggestion. In the revised manuscript, in addition to QUARK we now made comparisons of C-QUARK with three contact/distance-based folding methods, including CNS, DConStruct and trRosetta (using only contact as restraints). Overall, the results show that C-QUARK significantly outperforms any of these state-of-the-art control methods. These results are summarized in **Fig. 4** and **Fig. S9**, and **Tables S5-S7**, and discussed in Page 6-8 of **Main Text**:

To further quantitatively examine the importance of the comprehensive force field, we compared the performance of C-QUARK with three other programs that build structural models mainly based on predicted contacts or distances, including CNS⁴⁴, DConStruct⁴⁵ and trRosetta⁷. Here, CNS constructs protein structures primarily based on the satisfaction of distance geometries. The DConStruct algorithm is similar to CNS, but also considers idealized secondary structure geometries and produces models using the Limited-memory Broyden-Fletcher-Goldfarb-Shanno⁴⁶ (L-BFGS) procedure found in the MODELLER package⁴⁷. trRosetta builds the model with two steps. The first is on L-BFGS energy minimization with a restrained version of Rosetta, where the restraints contain inter-residue distance and orientation distributions from deep residual neural network predictions. In the second step, statistical energy functions are added to the force field to relax the model.

Here, we implement CNS through the CONFOLD package⁴⁸. The input features for CNS and DConStruct are built on the same set of contact and secondary structure predictions as what are used in C-QUARK. Since trRosetta generates restraints on its own, we provided the same MSAs but used only the contact restraints (i.e., distances where the peak of the predicted distance distribution was lower than 8Å or the sum of probabilities below 8Å was greater than 0.5), to provide a fair comparison with C-QUARK.

The modeling results of C-QUARK, CNS and DConStruct on the 247 test proteins are summarized in Table S5, where the average TM-score of the first models by C-QUARK (0.606) was 14% and 16% higher than that of CNS (0.530) and DConStruct (0.524), respectively; the differences corresponded to p-values of 3.5×10^{-20} and 1.5×10^{-25} in Student's t-tests. Figs. S9A and S9B present a head-to-head TM-score comparison between the methods, where the first models from C-QUARK had a higher TM-score than CNS (DConStruct) in 199 (198) out of the 247 cases, while the CNS (DConStruct) models did so for only 48 (49) of the cases. Notably, out of the 59 targets which had either a low N_f (<15) or a low contact-map accuracy (<30%), C-QUARK generated correct folds for 24 cases (i.e., 41% of the cases), while CNS (DConStruct) obtained correct folds for only 4 (4) of the cases (Table S6). Since contact prediction with low N_f MSAs has been an outstanding bottleneck in contact-guided *ab initio* modeling¹¹, such a significantly increased success rate of C-QUARK in generating correct models for these challenging targets is particularly encouraging. Meanwhile, the TM-score of C-QUARK (0.428) for these 59 targets was also significantly (p-value= 1.36×10^{-6}) higher than that of QUARK (0.348), showing that contact-map predictions are still helpful for folding despite the relatively lower accuracy (Fig. S9C and Table S6).

Since the same contact-maps were used by all three programs, it is of interest to examine why C-QUARK could create models with obviously better quality, particularly for the cases with low N_f and low contact prediction accuracy. Here, we used models produced by C-QUARK and CNS to highlight the reasons. Figs. 4A and 4D show an example from 3teqB, an *alpha*-protein packed with two anti-parallel, long helices. The N_f value for this target was relatively low (=12.2), which resulted in the contact-map (red circles in Fig. 4D) being comprised of many falsely predicted contacts. Overall, the contact prediction accuracy was 0.273 and 0.213 for long- and all-range contacts, respectively. With the help of the SSE prediction and pair-wise atomic and helix packing interactions contained in the inherent C-QUARK force field, C-QUARK eliminated the majority of the false-positive contacts during the simulations, as observed in the contact-map of the final model in Fig. 4D (blue circles in the left triangle) with accuracies of 0.667 and 0.500 for long- and all-range contacts, respectively. As a result, C-QUARK generated a model with a similar fold to the native of a TM-score of 0.658, shown in blue in Fig. 4A. On the other hand, the helices in the CNS model (shown in green in Fig. 4A) were bent in an unrealistic fashion due to the satisfaction of false-positive contacts (blue circles in the right triangle of Fig. 4D), resulting in a model with a low TM-score (0.289). It is noted that without contact information, C-QUARK would not be able to obtain a correct model as the TM-score of the QUARK model was only 0.44 for this target, demonstrating again the importance of the complementarity of the QUARK force field and the contact restraints even at a low accuracy.

Figs. 4B and 4E show another example from 1zuaA, which is a small *beta*-protein with 56 residues. Here, the N_f was very high (=1504.9), and hence the contact prediction accuracy for short-, medium, long- and all-range contacts was relatively high with accuracies of 0.6, 0.625, 0.659 and 0.627, respectively. The accuracies of the contact-maps derived from the final C-QUARK models increased further to 0.897, 0.836, 0.775 and 0.831, respectively, due to the removal of false positive contacts that clashed with the pairwise atomic interactions and hydrogen bonding between the *beta*-strands that formed the *beta*-sheets. As a result, the TM-score of the C-QUARK model for this target was 0.808. On the other hand, the TM-score of the CNS model was only 0.271, mainly due to false-positive contacts (highlighted by the dashed circles in Fig. 4E) that were correctly filtered out by C-QUARK but that incorrectly guided the CNS modeling.

One of the hallmarks of C-QUARK is that even if contact restraints are not present for some region of the query, the inherent QUARK potential can often help compensate for their absence and create correct full-length models. Fig. 4C shows such an example from 4yy2A, for which the native contacts between the N- and C-terminal helices (H_N and H_C) were not predicted (i.e., the red circles are largely absent in the rectangles in Fig. 4F). Additionally, due to the low N_f ($=0.402$), numerous false positive contacts were scattered around the contact-map. Despite the lack of contacts in the helix regions and the use of noisy contact restraints, the inherent QUARK potential correctly captured the interaction of the terminal helices and generated a model with a correct fold and a high TM-score of 0.813. On the other hand, CNS generated a completely wrong model with a TM-score= 0.290 by satisfying too many of the false positive contacts. In particular, due to the missing H_N - H_C contact restraints, the N-terminal helix was positioned far away from the C-terminal helix in the CNS model.

It is important to note that in the construction of our test dataset, homologous entries with sequence identities $>30\%$ to the training proteins of C-QUARK were filtered out. However, sequences homologous to the training sets of ResPRE and other third-party contact predictors, whose contact predictions are used by C-QUARK, were not particularly excluded from our test dataset. One reason is that the training sets for contact predictors are very large (e.g., the ResPRE training set included about 5,600 high-resolution protein structures and DeepContact utilized around 14,000 proteins from SCOPe 2.06 to train the method, etc.), to facilitate effective deep-learning training. Thus, the filtering of homologous proteins from these training sets would result in an insufficient number of proteins in the test dataset. Furthermore, C-QUARK, CNS and DConStruct utilized the same set of contacts, thus we did not specifically filter out the homologous proteins in the test set. However, since trRosetta generates spatial restraints using its own deep-learning predictor, to provide a fair comparison, we constructed a new test dataset by removing proteins with a 50% sequence identity to not only the training sets of all the contact predictors used by C-QUARK, but also the training set of trRosetta. This resulted in only 57 proteins being left in our new test dataset. Table S7 shows the results for the modeling performance of C-QUARK, CNS, DConStruct and trRosetta on this reduced test set. The TM-score of the C-QUARK models on this reduced dataset was slightly lower than that of the entire test set (compared to Table S6), probably due to the fact that this sub-dataset is non-redundant with the training set thus more difficult for contact prediction as the average accuracy was also reduced for CNS and DConStruct. Nevertheless, C-QUARK still significantly outperformed all the other control methods on this reduced dataset. It is notable that C-QUARK was 13.4% better than trRosetta, which was modified to only use predicted contacts derived from the distance predictions as restraints, in terms of the average TM-score of the first models (Fig S9D). Despite the relax step of trRosetta also used physical and knowledge-based potential, the global fold was primarily decided by the energy minimization step that only used predicted restraints. These results again demonstrate that C-QUARK outperforms other contact-based folding programs, mainly due to the help from its comprehensive force field used in the structural assembly simulations.

We also provided more specific comparisons between C-QUARK with RaptorX-DeepModeller, RaptorX-Contact, RaptorX-TBM, BAKER-ROSETTASERVER and Zhou-SPOT-3D servers on 64 CASP13 targets, results of which are summarized in Table S8. The revised text is in page 8-9 of Main Text.

Performance of C-QUARK on CASP13 targets

Not all methods have standalone packages available. To compare C-QUARK directly with other state-of-the-art structure prediction programs, C-QUARK participated in the 13th Critical Assessment of Structure Prediction (CASP13) experiment as “QUARK” server. Here, we analyzed the performance of C-QUARK on the 64 CASP13 FM, FM/TBM and TBM-hard targets (Dataset S3 in SI). By definition, these targets are

challenging since homologous templates are absent or difficult to detect from the PDB library. **Table S8** lists the average TM-scores and GDT_TS scores of the first predicted models by C-QUARK and the other best five server groups in the CASP13 experiment. Here, GDT_TS is the standard score metric used by the CASP assessors. We collected the models directly from the “QUARK” server which ran C-QUARK.

This dataset should provide a relatively fairer test with state-of-the-art structure modeling programs since most programs used sequence-based contact-maps and all the programs had access to the most recent sequence databases released before CASP13. Based on the experimental structures of 64 CASP13 targets, the average GDT_TS of C-QUARK was higher than that of all other participating servers with p-values <0.05 as calculated by Student’s t-tests (**Table S8**). Especially in the TBM-hard and FM categories, C-QUARK was 4% and 5% better than the second-best method, respectively. For FM/TBM targets, BAKER-ROSETTASERVER (60.58) was slightly better than C-QUARK (58.94), but the difference was not statistically significant. Some programs, such as the RaptorX servers, also used sequence-based distance map predictions⁵⁰. It is notable that RaptorX-Contact predicted the residue-residue distances, and then fed the restraints into CNS to reconstruct the 3D models. The average GDT_TS score of the C-QUARK first models (52.09) was still 12% higher than that of the RaptorX-Contact server (46.56). This gap was slightly smaller than the difference between C-QUARK and CNS in our benchmark test set (**Table S5**, where C-QUARK was 16% better than CNS), which is probably because, compared to contact prediction, additional information can be extracted from distance predictions to help guide the CNS modeling in RaptorX-contact. In **Fig. S12**, we highlight one of the FM targets, T0980s1-D1, which contains 105 residues with a 5-strand fold packed with an opposite helix. The TM-score of the first C-QUARK model was 0.540 for this domain, while the models generated by all other servers had TM-scores below 0.5. The poorer prediction for this target by other programs may be partially attributed to a low N_f value (8.2) and, subsequently, low accuracy in predicted contacts with numerous false positives (highlighted by the rectangles in **Fig. S12B**). On the other hand, most of the falsely predicted contacts were avoided in the C-QUARK models, due to the complementary effect of the inherent knowledge-based force field and the fragment-based distance profiles, which helped to correctly fold this target.

Response to Reviewer #2

We very much appreciate the comments and suggestions from the Reviewer, which points out multiple unclear places of the current manuscript, that help to significantly improve the quality and description of the manuscript. The major concerns from this Reviewer are on (i) the way of benchmark test set construction; (ii) the unclear roles played by fragment modules in C-QUARK; (iii) lack of comparisons with some of the most state-of-the-art methods; (iv) outdated results from early CASP datasets; (v) lack of rational of 3D contact potentials; and (vi) lack of results on multi-domain proteins. We carefully addressed the Reviewer's comments and suggestions by adding and discussing additional experimental data in the revised manuscript. Below, we include point-by-point replies to the comments of the Reviewer, where all changes have been highlighted in yellow in the manuscript.

1. **The Reviewer commented:**

The manuscript by S. M. Mortuza et al. describes a new folding method C-QUARK for protein structure prediction. The method is a significant extension of QUARK, an excellent software in the field. C-QUARK extends the QUARK force field with predicted contact map and then uses the extended force field to guide Monte Carlo fragment assembly simulations. One of the key elements of C-QUARK is 3G contact potential that selects contacts predicted by multiple programs. Experimental results suggest that C-QUARK outperforms QUARK and CNS in ab initio protein structure prediction, especially for hard targets, i.e., the beta/alpha-beta proteins with complicated topologies or low Nf value. Another advantage of C-QUARK is its robustness to the falsely predicted contacts (and even corrects the false-positive contacts), which is mainly due to the complementation between contacts and the QUARK force field. I also appreciate the failure analysis of C-QUARK, which is very interesting and instructive for further improvement.

We appreciate the positive comments of the Reviewer on the work, and the nice summary of the strength of the C-QUARK algorithm.

2. **The Reviewer commented:**

Major comments:

1. The comparison is not enough to show the advantage of C-QUARK over other contact/distance-based methods. The authors shall compare their method with advanced contact-based approaches, such as AlphaFold and trRosetta.

We thank the Reviewer's suggestion. Following this, we first added a comparison between C-QUARK, AlphaFold and trRosetta, which focused on 64 CASP13 targets since there is no standalone package or server for AlphaFold. Since trRosetta did not participate in CASP13 (developed after CASP13), we run trRosetta standalone package using the default setting with the same MSA that we used for C-QUARK during CASP13. Here, we were unable to use the MSAs from the trRosetta paper, as we could not obtain the scripts or databases of trRosetta for MSA generation, and the MSAs provided on the trRosetta website are only for 25 CASP targets, furthermore their MSA collected from a sequence database constructed after CASP13.

The comparison results between C-QUARK and AlphaFold and trRosetta are summarized in **Table S9**. Overall, although C-QUARK outperformed trRosetta and AlphaFold for the TBM-hard targets, the TM-score of C-QUARK was lower than the latter two on FM/TBM and FM targets, which resulted in a lower average TM-score on all 64 targets. These results are discussed in page 10 of the **Main Text**:

Third, although C-QUARK outperformed most of the servers, including those using distance restraints, in CASP13, the most recent progress of the field showed advancement in modeling accuracy using deep-learning distance, inter-residue torsion angle and hydrogen bonding restraints for *ab initio* structure predictions^{6, 7, 31, 52}. Due to the limited information provided by the binary distance classification in contact prediction, folding programs that solely use contact restraints may not be comparable with the most advanced programs that combine contact restraints with those new categories of spatial restraints (see the comparisons between C-QUARK, AlphaFold and trRosetta in **Table S9** on the 64 CASP13 FM targets).

Table S9: The average TM-scores and GDT_TS scores of the first models produced by C-QUARK, AlphaFold and trRosetta on the CASP13 targets in the FM, FM/TBM and TBM-hard categories. The values in the parentheses are the p-values calculated by Student’s t-tests between C-QUARK and the other control programs.

Target type	Methods	Average TM-score	Average GDT_TS
All (64 targets)	C-QUARK	0.588	52.09
	AlphaFold	0.648 ($1.00 \times 10^{+0}$)	58.43 ($1.00 \times 10^{+0}$)
	trRosetta	0.619 (9.99×10^{-1})	55.34 (9.97×10^{-1})
TBM-hard (21 targets)	C-QUARK	0.720	61.03
	AlphaFold	0.710 (5.41×10^{-1})	61.80 (5.90×10^{-1})
	trRosetta	0.680 (1.92×10^{-1})	57.93 (1.64×10^{-1})
FM/TBM (12 targets)	C-QUARK	0.598	58.94
	AlphaFold	0.695 (9.98×10^{-1})	68.22 (9.89×10^{-1})
	trRosetta	0.622 (7.50×10^{-1})	61.56 (8.03×10^{-1})
FM (31 targets)	C-QUARK	0.495	43.38
	AlphaFold	0.589 ($1.00 \times 10^{+0}$)	52.35 ($1.00 \times 10^{+0}$)
	trRosetta	0.577 ($1.00 \times 10^{+0}$)	51.17 ($1.00 \times 10^{+0}$)

This result is understandable because both AlphaFold and trRosetta (developed after CASP13) used distance prediction instead of contact prediction as restraints and predicted distances can provide additional information beyond the contact prediction. Additionally, AlphaFold is a human group in CASP13 while C-QUARK is an automated server group. The human group has 21 days for modeling one protein while server group only 72 hours. Consider the resources used by AlphaFold, these may not be an entirely fair comparison.

Nevertheless, given the special role of contact-map prediction in protein folding and the fact that most of the predicted distances and orientations are on the residue pairs with short distance (i.e., in contact), we believe it is of critical importance to study and benchmark separately the impact of contact-maps on the problem of *ab initio* protein structure prediction, and systematically examine the critical weakness and strength of deep-learning

contact restraints when coupled with the advanced protein folding simulation algorithms. Thus, we further compared C-QUARK with trRosetta that only used contact-like distances, i.e., the distance restraints with a peak of predicted distance distribution lower than 8Å or the sum probabilities below 8Å is greater than 0.5. Furthermore, we make a comparison with other two contact-based folding programs, CNS in CONFOLD package and DConStruct. Overall, C-QUARK outperforms all three contact-based modeling methods, showing the advantage of optimized combination of contact restraints with the advanced folding simulations in C-QUARK. These results are summarized in **Fig. 4** and **Fig. S9**, and **Tables S5-S7**, and discussed in Page 6-8 of **Main Text**:

To further quantitatively examine the importance of the comprehensive force field, we compared the performance of C-QUARK with three other programs that build structural models mainly based on predicted contacts or distances, including CNS⁴⁴, DConStruct⁴⁵ and trRosetta⁷. Here, CNS constructs protein structures primarily based on the satisfaction of distance geometries. The DConStruct algorithm is similar to CNS, but also considers idealized secondary structure geometries and produces models using the Limited-memory Broyden–Fletcher–Goldfarb–Shanno⁴⁶ (L-BFGS) procedure found in the MODELLER package⁴⁷. trRosetta builds the model with two steps. The first is on L-BFGS energy minimization with a restrained version of Rosetta, where the restraints contain inter-residue distance and orientation distributions from deep residual neural network predictions. In the second step, statistical energy functions are added to the force field to relax the model. Here, we implement CNS through the CONFOLD package⁴⁸. The input features for CNS and DConStruct are built on the same set of contact and secondary structure predictions as what are used in C-QUARK. Since trRosetta generates restraints on its own, we provided the same MSAs but used only the contact restraints (i.e., distances where the peak of the predicted distance distribution was lower than 8Å or the sum of probabilities below 8Å was greater than 0.5), to provide a fair comparison with C-QUARK.

The modeling results of C-QUARK, CNS and DConStruct on the 247 test proteins are summarized in **Table S5**, where the average TM-score of the first models by C-QUARK (0.606) was 14% and 16% higher than that of CNS (0.530) and DConStruct (0.524), respectively; the differences corresponded to p-values of 3.5×10^{-20} and 1.5×10^{-25} in Student's t-tests. **Figs. S9A** and **S9B** present a head-to-head TM-score comparison between the methods, where the first models from C-QUARK had a higher TM-score than CNS (DConStruct) in 199 (198) out of the 247 cases, while the CNS (DConStruct) models did so for only 48 (49) of the cases. Notably, out of the 59 targets which had either a low N_f (<15) or a low contact-map accuracy (<30%), C-QUARK generated correct folds for 24 cases (i.e., 41% of the cases), while CNS (DConStruct) obtained correct folds for only 4 (4) of the cases (**Table S6**). Since contact prediction with low N_f MSAs has been an outstanding bottleneck in contact-guided *ab initio* modeling¹¹, such a significantly increased success rate of C-QUARK in generating correct models for these challenging targets is particularly encouraging. Meanwhile, the TM-score of C-QUARK (0.428) for these 59 targets was also significantly (p-value= 1.36×10^{-6}) higher than that of QUARK (0.348), showing that contact-map predictions are still helpful for folding despite the relatively lower accuracy (**Fig. S9C** and **Table S6**).

Since the same contact-maps were used by all three programs, it is of interest to examine why C-QUARK could create models with obviously better quality, particularly for the cases with low N_f and low contact prediction accuracy. Here, we used models produced by C-QUARK and CNS to highlight the reasons. **Figs. 4A** and **4D** show an example from 3teqB, an *alpha*-protein packed with two anti-parallel, long helices. The N_f value for this target was relatively low (=12.2), which resulted in the contact-map (red circles in **Fig. 4D**) being comprised of many falsely predicted contacts. Overall, the contact prediction accuracy was 0.273 and 0.213 for long- and all-range contacts, respectively. With the help of the SSE prediction and pair-wise atomic and helix packing interactions contained in the inherent C-QUARK force field, C-QUARK eliminated the majority of the

false-positive contacts during the simulations, as observed in the contact-map of the final model in **Fig. 4D** (blue circles in the left triangle) with accuracies of 0.667 and 0.500 for long- and all-range contacts, respectively. As a result, C-QUARK generated a model with a similar fold to the native of a TM-score of 0.658, shown in blue in **Fig. 4A**. On the other hand, the helices in the CNS model (shown in green in **Fig. 4A**) were bent in an unrealistic fashion due to the satisfaction of false-positive contacts (blue circles in the right triangle of **Fig. 4D**), resulting in a model with a low TM-score (0.289). It is noted that without contact information, C-QUARK would not be able to obtain a correct model as the TM-score of the QUARK model was only 0.44 for this target, demonstrating again the importance of the complementarity of the QUARK force field and the contact restraints even at a low accuracy.

Figs. 4B and **4E** show another example from 1zuaA, which is a small *beta*-protein with 56 residues. Here, the N_f was very high (=1504.9), and hence the contact prediction accuracy for short-, medium, long- and all-range contacts was relatively high with accuracies of 0.6, 0.625, 0.659 and 0.627, respectively. The accuracies of the contact-maps derived from the final C-QUARK models increased further to 0.897, 0.836, 0.775 and 0.831, respectively, due to the removal of false positive contacts that clashed with the pairwise atomic interactions and hydrogen bonding between the *beta*-strands that formed the *beta*-sheets. As a result, the TM-score of the C-QUARK model for this target was 0.808. On the other hand, the TM-score of the CNS model was only 0.271, mainly due to false-positive contacts (highlighted by the dashed circles in **Fig. 4E**) that were correctly filtered out by C-QUARK but that incorrectly guided the CNS modeling.

One of the hallmarks of C-QUARK is that even if contact restraints are not present for some region of the query, the inherent QUARK potential can often help compensate for their absence and create correct full-length models. **Fig. 4C** shows such an example from 4yy2A, for which the native contacts between the N- and C-terminal helices (H_N and H_C) were not predicted (i.e., the red circles are largely absent in the rectangles in **Fig. 4F**). Additionally, due to the low N_f (=0.402), numerous false positive contacts were scattered around the contact-map. Despite the lack of contacts in the helix regions and the use of noisy contact restraints, the inherent QUARK potential correctly captured the interaction of the terminal helices and generated a model with a correct fold and a high TM-score of 0.813. On the other hand, CNS generated a completely wrong model with a TM-score=0.290 by satisfying too many of the false positive contacts. In particular, due to the missing H_N - H_C contact restraints, the N-terminal helix was positioned far away from the C-terminal helix in the CNS model.

It is important to note that in the construction of our test dataset, homologous entries with sequence identities >30% to the training proteins of C-QUARK were filtered out. However, sequences homologous to the training sets of ResPRE and other third-party contact predictors, whose contact predictions are used by C-QUARK, were not particularly excluded from our test dataset. One reason is that the training sets for contact predictors are very large (e.g., the ResPRE training set included about 5,600 high-resolution protein structures and DeepContact utilized around 14,000 proteins from SCOPe 2.06 to train the method, etc.), to facilitate effective deep-learning training. Thus, the filtering of homologous proteins from these training sets would result in an insufficient number of proteins in the test dataset. Furthermore, C-QUARK, CNS and DConStruct utilized the same set of contacts, thus we did not specifically filter out the homologous proteins in the test set. However, since trRosetta generates spatial restraints using its own deep-learning predictor, to provide a fair comparison, we constructed a new test dataset by removing proteins with a 50% sequence identity to not only the training sets of all the contact predictors used by C-QUARK, but also the training set of trRosetta. This resulted in only 57 proteins being left in our new test dataset. **Table S7** shows the results for the modeling performance of C-QUARK, CNS, DConStruct and trRosetta on this reduced test set. The TM-score of the C-QUARK models on this reduced dataset was slightly lower than that of the entire test set (compared to **Table S6**), probably due to the fact that this sub-dataset is non-redundant with the training set thus more difficult for contact prediction as the average accuracy was also reduced for CNS and DConStruct. Nevertheless, C-QUARK

still significantly outperformed all the other control methods on this reduced dataset. It is notable that C-QUARK was 13.4% better than trRosetta, which was modified to only use predicted contacts derived from the distance predictions as restraints, in terms of the average TM-score of the first models (**Fig S9D**). Despite the relax step of trRosetta also used physical and knowledge-based potential, the global fold was primarily decided by the energy minimization step that only used predicted restraints. These results again demonstrate that C-QUARK outperforms other contact-based folding programs, mainly due to the help from its comprehensive force field used in the structural assembly simulations.

3. The Reviewer commented:

2 The construction of the benchmark dataset (lines 95-97) is not very clear. Are these 247 test proteins collected in a fair way? Are they already in the training set of the used contact predictors (e.g. ResPRE and DeepCov)? The authors shall clearly indicate the potential overlap between test and training sets for all used contact predictors.

The Reviewer raised an important question. In the construction of our testing dataset, the homologous entries with sequence identities >30% to the training proteins of C-QUARK were filtered out. However, sequences homologous to the proteins in the ResPRE and other contact predictor training set, whose contact predictions are used by C-QUARK, were not particularly excluded from our testing datasets. One reason is that the training sets for contact predictor are very large (e.g., the ResPRE training set included about 5,600 high-resolution protein structures to facilitate effective deep-learning training, the DeepContact utilized around 14,000 proteins from SCOPe 2.06 to train the method, etc.). Thus, the filtering of homologous proteins from those training set would result in an insufficient number of proteins in the testing dataset.

To partially address this issue, we collected a new testing dataset by removing proteins with a sequence identity <50% to not only the training sets for all contact predictors used by C-QUARK, but also the training set of trRosetta, which resulted in only 57 proteins left. The results are shown in **Table S7** in **SI**. Overall, C-QUARK outperforms all these contact-based folding methods in our comparison based on this set of non-redundancy test proteins. We clarify this problem in page 7 of the **Main Text**:

It is important to note that in the construction of our test dataset, homologous entries with sequence identities >30% to the training proteins of C-QUARK were filtered out. However, sequences homologous to the training sets of ResPRE and other third-party contact predictors, whose contact predictions are used by C-QUARK, were not particularly excluded from our test dataset. One reason is that the training sets for contact predictors are very large (e.g., the ResPRE training set included about 5,600 high-resolution protein structures and DeepContact utilized around 14,000 proteins from SCOPe 2.06 to train the method, etc.), to facilitate effective deep-learning training. Thus, the filtering of homologous proteins from these training sets would result in an insufficient number of proteins in the test dataset. Furthermore, C-QUARK, CNS and DConStruct utilized the same set of contacts, thus we did not specifically filter out the homologous proteins in the test set. However, since trRosetta generates spatial restraints using its own deep-learning predictor, to provide a fair comparison, we constructed a new test dataset by removing proteins with a 50% sequence identity to not only the training sets of all the contact predictors used by C-QUARK, but also the training set of trRosetta. This resulted in only

57 proteins being left in our new test dataset. **Table S7** shows the results for the modeling performance of C-QUARK, CNS, DConStruct and trRosetta on this reduced test set. The TM-score of the C-QUARK models on this reduced dataset was slightly lower than that of the entire test set (compared to **Table S6**), probably due to the fact that this sub-dataset is non-redundant with the training set thus more difficult for contact prediction as the average accuracy was also reduced for CNS and DConStruct. Nevertheless, C-QUARK still significantly outperformed all the other control methods on this reduced dataset. It is notable that C-QUARK was 13.4% better than trRosetta, which was modified to only use predicted contacts derived from the distance predictions as restraints, in terms of the average TM-score of the first models (**Fig S9D**). Despite the relax step of trRosetta also used physical and knowledge-based potential, the global fold was primarily decided by the energy minimization step that only used predicted restraints. These results again demonstrate that C-QUARK outperforms other contact-based folding programs, mainly due to the help from its comprehensive force field used in the structural assembly simulations.

Table S7: Average TM-scores, GDT_TS scores and RMSDs for the first models generated by C-QUARK, QUARK, CNS, DConStruct and trRosetta on the 57 targets of the test set without redundancy to the trRosetta training set and all training sets of the contact predictors used by C-QUARK. Here, trRosetta used only the contact restraints, i.e., distances where the peak of the predicted distance distribution was lower than 8Å or the sum of probabilities below 8Å was greater than 0.5, to provide a fair comparison with C-QUARK. The values in the parentheses of the second, third and fourth columns represent the p-values based on Student's t-tests. Additionally, the values in parentheses of the fifth column represent the percentage of the cases where the models obtained similar folds as the corresponding native structures.

Method	TM-score	GDT_TS	RMSD	Number of cases with TM-score \geq 0.5
C-QUARK	0.525	48.19	8.64	30 (53%)
QUARK	0.418 (8.63×10^{-7})	39.19 (3.84×10^{-6})	13.09 (6.28×10^{-7})	17 (30%)
CNS	0.440 (2.61×10^{-9})	39.70 (4.34×10^{-9})	10.48 (2.70×10^{-6})	21 (37%)
DConStruct	0.438 (3.64×10^{-9})	39.07 (2.38×10^{-9})	9.95 (6.83×10^{-4})	21 (37%)
trRosetta (contact)	0.463 (3.96×10^{-2})	42.31 (5.43×10^{-2})	10.47 (4.91×10^{-3})	20 (35%)

In addition, in our comparison with CNS and DConStruct on the 247 test proteins, the same set of contacts was utilized in C-QUARK, CNS and DConStruct. Therefore, this part of results was not related with whether homology proteins from the training set of contact predictors are filtered or not. Since trRosetta needs to use the restraints generated by its own format of deep-learning predictions, it was not included in the tests on the 247 proteins.

4. The Reviewer commented:

3 The performance for free-modeling targets is ambiguous. The authors shall evaluate CASP13 FM, FM/TBM, and TBM-hard targets separately.

Thank you for Reviewer's suggestion. Following this, we separately evaluated the CASP13 targets based on FM, FM/TBM, TBM-hard and All (combine all three categories) targets, with results summarized in **Tables S8**. Here, we remove three server-only targets in **Table S8**, in order to make the dataset being the same with **Table S9** in which the CASP13 human

group (AlphaFold) was compared with C-QUARK. Thus, the average value and p-value could change accordingly compared to the **SI** in the last version.

Overall, C-QUARK outperforms all other servers in All, TBM-hard and FM categories of CASP13, but slightly (or statistically insignificantly) worse than BAKER-ROSETTASERVER in TBM/FM targets. These results are discussed in the following paragraphs (Page 8-9):

Performance of C-QUARK on CASP13 targets

Not all methods have standalone packages available. To compare C-QUARK directly with other state-of-the-art structure prediction programs, C-QUARK participated in the 13th Critical Assessment of Structure Prediction (CASP13) experiment as “QUARK” server. Here, we analyzed the performance of C-QUARK on the 64 CASP13 FM, FM/TBM and TBM-hard targets (**Dataset S3** in **SI**). By definition, these targets are challenging since homologous templates are absent or difficult to detect from the PDB library. **Table S8** lists the average TM-scores and GDT_TS scores of the first predicted models by C-QUARK and the other best five server groups in the CASP13 experiment. Here, GDT_TS is the standard score metric used by the CASP assessors. We collected the models directly from the “QUARK” server which ran C-QUARK.

This dataset should provide a relatively fairer test with state-of-the-art structure modeling programs since most programs used sequence-based contact-maps and all the programs had access to the most recent sequence databases released before CASP13. Based on the experimental structures of 64 CASP13 targets, the average GDT_TS of C-QUARK was higher than that of all other participating servers with p-values <0.05 as calculated by Student’s t-tests (**Table S8**). Especially in the TBM-hard and FM categories, C-QUARK was 4% and 5% better than the second-best method, respectively. For FM/TBM targets, BAKER-ROSETTASERVER (60.58) was slightly better than C-QUARK (58.94), but the difference was not statistically significant. Some programs, such as the RaptorX servers, also used sequence-based distance map predictions⁵⁰. It is notable that RaptorX-Contact predicted the residue-residue distances, and then fed the restraints into CNS to reconstruct the 3D models. The average GDT_TS score of the C-QUARK first models (52.09) was still 12% higher than that of the RaptorX-Contact server (46.56). This gap was slightly smaller than the difference between C-QUARK and CNS in our benchmark test set (**Table S5**, where C-QUARK was 16% better than CNS), which is probably because, compared to contact prediction, additional information can be extracted from distance predictions to help guide the CNS modeling in RaptorX-contact. In **Fig. S12**, we highlight one of the FM targets, T0980s1-D1, which contains 105 residues with a 5-strand fold packed with an opposite helix. The TM-score of the first C-QUARK model was 0.540 for this domain, while the models generated by all other servers had TM-scores below 0.5. The poorer prediction for this target by other programs may be partially attributed to a low N_f value (8.2) and, subsequently, low accuracy in predicted contacts with numerous false positives (highlighted by the rectangles in **Fig. S12B**). On the other hand, most of the falsely predicted contacts were avoided in the C-QUARK models, due to the complementary effect of the inherent knowledge-based force field and the fragment-based distance profiles, which helped to correctly fold this target.

Table S8: The average TM-scores and GDT_TS scores of the first models by C-QUARK on the CASP targets in comparison to the top five servers on FM, FM/TBM and TBM-hard targets in CASP13. The values in the parentheses are the p-values calculated by Student’s t-tests between C-QUARK and the other control programs. We did not show ‘Zhang-Server’ in CASP13 because it used C-QUARK models as the starting models for FM targets.

Target type	Methods	Average TM-score	Average GDT_TS
-------------	---------	------------------	----------------

All (64 targets)	C-QUARK (participated as “QUARK”)	0.588	52.09
	RaptorX-DeepModeller	0.558 (2.24×10^{-2})	49.38 (1.89×10^{-2})
	RaptorX-Contact	0.531 (3.31×10^{-4})	46.56 (8.90×10^{-5})
	RaptorX-TBM	0.521 (1.94×10^{-6})	45.92 (2.99×10^{-6})
	BAKER-ROSETTASERVER	0.513 (2.47×10^{-4})	45.76 (6.86×10^{-4})
	Zhou-SPOT-3D	0.447 (1.15×10^{-9})	38.77 (7.61×10^{-10})
TBM-hard (21 targets)	C-QUARK (participated as “QUARK”)	0.720	61.03
	RaptorX-DeepModeller	0.682 (7.35×10^{-2})	58.04 (3.58×10^{-2})
	RaptorX-Contact	0.613 (5.88×10^{-4})	50.97 (3.60×10^{-4})
	RaptorX-TBM	0.686 (8.39×10^{-2})	58.11 (3.55×10^{-2})
	BAKER-ROSETTASERVER	0.644 (1.96×10^{-1})	54.69 (2.47×10^{-1})
	Zhou-SPOT-3D	0.576 (3.04×10^{-4})	46.40 (1.09×10^{-3})
FM/TBM (12 targets)	C-QUARK (participated as “QUARK”)	0.598	58.94
	RaptorX-DeepModeller	0.572 (3.39×10^{-1})	56.45 (1.79×10^{-1})
	RaptorX-Contact	0.525 (4.47×10^{-2})	51.54 (1.78×10^{-2})
	RaptorX-TBM	0.538 (1.05×10^{-2})	53.21 (2.89×10^{-2})
	BAKER-ROSETTASERVER	0.609 (6.54×10^{-1})	60.58 (7.01×10^{-1})
	Zhou-SPOT-3D	0.489 (3.36×10^{-2})	48.91 (2.88×10^{-2})
FM (31 targets)	C-QUARK (participated as “QUARK”)	0.495	43.38
	RaptorX-DeepModeller	0.468 (1.32×10^{-1})	40.79 (9.62×10^{-2})
	RaptorX-Contact	0.477 (1.60×10^{-1})	41.64 (1.51×10^{-1})
	RaptorX-TBM	0.402 (1.23×10^{-4})	34.84 (1.24×10^{-4})
	BAKER-ROSETTASERVER	0.388 (5.92×10^{-5})	33.98 (8.85×10^{-5})
	Zhou-SPOT-3D	0.343 (7.87×10^{-9})	29.68 (1.07×10^{-7})

5. The Reviewer commented:

4. The comparisons over CASP10-12 targets are not very fair. The authors use newer sequence and structure databases, thus obtain significantly better contacts than other predictors. It is recommended to remove these comparisons. By the way, are these CASP10-12 targets excluded from the training set for all used contact predictors?

We agree with the Reviewer that the comparisons over CASP10-12 targets are unfair. Accordingly, we removed the CASP10-12 targets from manuscript, and only kept CASP13 targets for the comparison.

Multiple independent contact predictors, including those developed by the third-party groups, were used in C-QUARK. Some of them were developed after CASP12, and we do not believe that the CASP10-12 targets have been specifically excluded from the training set of these contact predictors.

6. The Reviewer commented:

5. It is interesting to know how fragments affect performance as fragments key parts of C-QUARK. Although fragment-assembly has been shown successfully in QUARK, it is

expected to show the necessity of fragments when high-quality contacts are given. I recommend the authors to carry out the following experiments:

a. To show the effects of fragments in the score function, the author might compare C-QUARK with a baseline model that discards all fragment-related energy items (e.g. fragment-based distance profile) in the C-QUARK force field.

b. To show the effects of fragments in the optimization method, the author might compare C-QUARK with a fragment-free optimization method, such as gradient descent (as the contact-map energy function is differentiable).

Thanks for the excellent suggestion, which should help clearly highlight the importance of fragments in C-QUARK. For suggestion (a), we built a baseline pipeline that removes the fragments-based energy potential (i.e., distance-profile energy term) from C-QUARK. For suggestion (b), since most of the energy terms of C-QUARK force field are not differentiable, implementing a gradient descent-based fragments-free optimization method is not feasible. Alternatively, we completely removed the fragment module from C-QUARK, including the fragments-based energy potential and fragments replacement movements in the Replica Exchange Monte Carlo assembly simulation stage. We compared the performance of C-QUARK and these two variant C-QUARK baseline pipelines on the 247 test proteins. Overall, both baseline methods, especially the baseline (b), have significantly worse performance than the complete C-QUARK pipeline. Those results show that the fragments module affects the performance of C-QUARK even when high-quality contacts are given.

The result of the comparison is summarized in **Table S4** in **SI**. We added the following paragraph to discuss the data in Page 6 of the **Main Text**:

While contact-map predictions greatly help in *ab initio* folding, other physical and knowledge-based energy terms, including pairwise atomic potentials, solvation, hydrogen bonding, secondary structure element (SSE) packing and fragment-based distance profile in C-QUARK (Eq. S2), also play important roles in improving modeling accuracy, e.g., by filtering out contacts that are physically unrealistic. Such complementarity between the contact potential and the inherent QUARK force field is vital in *ab initio* modeling. For instance, if the fragment-based distance-profile term is removed from the C-QUARK force field, the average TM-score of the first models by C-QUARK decreases from 0.606 to 0.593 with a p-value of 4.16×10^{-4} (Table S4). Furthermore, if the entire fragments module, including the fragment-profile energy term and the fragment replacement movements in the simulation optimization (see details in Methods), is excluded from C-QUARK, the performance will become much worse with TM-score reduced from 0.606 to 0.553 with a p-value of 1.59×10^{-30} . These data indicate that the structural fragment module plays an important role in C-QUARK, which further demonstrate that the success of C-QUARK should be attributed to the interplay of predicted residue-residue contacts and the inherent force field and structural assembly simulation process.

Table S4: Average TM-scores and GDT_TS scores (Global Distance Test Total Score) for the first models generated by C-QUARK, C-QUARK without the distance profile energy term, and C-QUARK without fragment-based optimization on the test set. The values in the parentheses of the second and third columns represent the p-values calculated by Student's t-tests. The values in parentheses in the fourth column represent the percentage of cases where the models obtained similar folds as the corresponding native structures. As per the CASP evaluation measurement, GDT_TS is calculated by $GDT_TS = (GDT_P1 + GDT_P2 + GDT_P4 + GDT_P8)/4$, where GDT_Pn denotes the percent of residues under the distance cut-off $\leq n$ Å.

Method	Average TM-score	Average GDT_TS	Number of cases with TM-score \geq 0.5
C-QUARK	0.606	53.90	186 (75%)
C-QUARK (no distance profile term)	0.593 (4.16×10^{-4})	52.51 (1.70×10^{-5})	184 (74%)
C-QUARK (no fragments)	0.553 (1.59×10^{-30})	48.41 (2.79×10^{-33})	162 (66%)

7. The Reviewer commented:

6. It is hard to judge the advantage of C-QUARK when lacking homologous sequences. For the 59 targets with low accuracy contacts (line 254-263), the authors compared C-QUARK with CNS. However, even if the authors have provided case studies, it is still not clear whether C-QUARK could perform better than QUARK with low-quality contacts. The authors shall explicitly show the performance of QUARK over these 59 targets.

Thank you for raising this issue. To address this issue, we listed in **Table S6** and **Figure S9C** the results of QUARK on the 59 targets, in comparison with C-QUARK. Although QUARK slightly outperforms CNS and DConStruct, demonstrating the advantage of fragment-assembly based simulations, C-QUARK still significantly outperformed QUARK. This data shows that the low-accuracy contact could still help C-QUARK modeling, probably due to the complementarity that can partly filter out the noise of the contact maps. We added the following paragraph to discuss the results (Page 6):

Notably, out of the 59 targets which had either a low N_f (<15) or a low contact-map accuracy ($<30\%$), C-QUARK generated correct folds for 24 cases (i.e., 41% of the cases), while CNS (DConStruct) obtained correct folds for only 4 (4) of the cases (**Table S6**). Since contact prediction with low N_f MSAs has been an outstanding bottleneck in contact-guided *ab initio* modeling¹¹, such a significantly increased success rate of C-QUARK in generating correct models for these challenging targets is particularly encouraging. Meanwhile, the TM-score of C-QUARK (0.428) for these 59 targets was also significantly (p-value= 1.36×10^{-6}) higher than that of QUARK (0.348), showing that contact-map predictions are still helpful for folding despite the relatively lower accuracy (**Fig. S9C** and **Table S6**).

Table S6: Average TM-scores, GDT_TS scores and RMSDs for the first models generated by C-QUARK, QUARK, CNS and DConStruct on the 59 targets of the test set with low contact-map prediction accuracy. The values in the parentheses of the second, third and fourth columns represent the p-values calculated by Student's t-tests. Additionally, the values in parentheses of the fifth column represent the percentage of the cases where the models obtained similar folds as the corresponding native structures.

Method	TM-score	GDT_TS	RMSD	Number of cases with TM-score \geq 0.5
C-QUARK	0.428	39.98	10.21	24 (41%)
QUARK	0.348 (1.36×10^{-6})	33.37 (6.53×10^{-6})	14.15 (1.13×10^{-6})	7 (12%)
CNS	0.324 (1.52×10^{-9})	30.25 (1.66×10^{-9})	12.65 (4.06×10^{-7})	4 (7%)
DConStruct	0.326 (3.02×10^{-9})	30.14 (3.46×10^{-9})	12.16 (6.59×10^{-5})	4 (7%)

Figure S9C: TM-score comparison between the first models produced by C-QUARK and QUARK for 59 test proteins with low accuracy of contact-map prediction. The dashed lines indicate the TM-score cut-off of 0.5, beyond which models are considered to obtain similar folds as the corresponding native structures. Points above the diagonal line indicate models with better quality by C-QUARK than the control methods, and vice versa.

8. The Reviewer commented:

7. The performance of CNS is inconsistent with that of RaptorX, which uses CNS to construct prediction models. The authors showed C-QUARK performs better than CNS (0.606 vs 0.530 in TM-score); however, C-QUARK didn't show such a superiority over RaptorX (51.396 vs 49.457 in GDT_TS). An in-depth examination of this issue is expected.

Thanks for raising the interesting question. To answer the question, we first add TM-score and GDT_TS to both benchmark (**Table S5**) and CASP (**Table S8**) targets, which let us make directly comparisons of the methods by either TM-score or GDT_TS. As shown in **Table S8**, there three versions of RaptorX servers. Based on the CASP13 Abstract book, the RaptorX-Contact server predicted the residue-residue distances and then fed the predicted distances into CNS as restraints to construct the 3D model. Thus, when comparing C-QUARK with the RaptorX server purely based on CNS, we should compare it with RaptorX-Contact server, instead of the RaptorX-DeepModeller server which is a meta-server combining the results from RaptorX-Contact and RaptorX-TBM (a distance-based threading server).

Based on **Table S8**, the GDT_TS score by C-QUARK and the RaptorX-Contact server that used CNS are 52.09 and 46.56, respectively (-please note that we removed three server-only targets in **Table S8** in order to facilitate the comparison with the human group of

AlphaFold in **Table S9**, thus the result is slightly different that from the former **SI**). This result is roughly consistent with the comparison of C-QUARK and CNS in the benchmark set which have GDT_TS of 53.90 and 46.51, respectively (**Table S5**). Here, the gap between C-QUARK and RaptorX-contact is still slightly smaller than that between C-QUARK and CNS. This might be due to the fact that RaptorX-contact uses distance restraints. Meanwhile, there are also variations in the number of testing targets and the contact map predictions between the benchmark test of CNS here and the RaptorX-Contact in CASP13.

We added the following paragraph to clarify the point in **Main Text** (Page 9):

Some programs, such as the RaptorX servers, also used sequence-based distance map predictions⁵⁰. It is notable that RaptorX-Contact predicted the residue-residue distances, and then fed the restraints into CNS to reconstruct the 3D models. The average GDT_TS score of the C-QUARK first models (52.09) was still 12% higher than that of the RaptorX-Contact server (46.56). This gap was slightly smaller than the difference between C-QUARK and CNS in our benchmark test set (**Table S5**, where C-QUARK was 16% better than CNS), which is probably because, compared to contact prediction, additional information can be extracted from distance predictions to help guide the CNS modeling in RaptorX-contact.

9. The Reviewer commented:

8. The 3G contact potential term (Eq. 1) is the key element of C-QUARK; however, this term seems ad hoc. What are the intuition and physics underlying this term?

Thank you for raising this question. Overall, as shown in **Fig. S17**, the 3G contact potential is centered with a negative well at 8 Å cutoff, with a strong force in 8 Å to D ($=8 \text{ \AA} + d_b$), followed by a weaker force in D to 80 Å, being introduced to push the target residue pairs towards the well when they are in a long distance.

The physical and intuitive consideration of the 3D contact potential is following. Since the contact prediction can only tell whether the distance between residue pair $i-j$ below 8 Å or not, we design the 3G potential as a flat well when distance $< 8 \text{ \AA}$. Since almost all of the residue-residue distance in normal size protein is lower than 80 Å, the potential is also designed as flat when distance is beyond the maximal distance threshold 80 Å. Between 8 Å and 80 Å, we split the potential into two regions by a dynamic threshold of D ($=8 \text{ \AA} + d_b$), with d_b changing from 6 to 12 Å depending on the target size (**Table S15**). In the region above D , a relatively weaker force is used to avoid structural overpacking due to false positive contact predictions, while in the region below D , a stronger force is used to push contact restraints quickly satisfied since in this region the contact accuracy of the target residue pairs is supposed to be higher than that in the longer-distance regions (-because most of the adjacent residue pairs in the structure decoys are supposed to be more consistent with the inherent QUARK potential after the equilibrium of Monte Carlo simulations). We selected trigonometric function style potential in these two regions, since trigonometric functions are the simple, continuous, smooth, and differentiable functions, which can easily make the connect points (8, D and 80 Å) differentiable.

We have revised the text in page 12 according to the Reviewer's comments.

Overall, the 3G potential contains a negative well at an 8 Å cutoff, with a strong force from 8 Å to $D (=8 \text{ \AA} + d_b)$, followed by a weaker force from D to 80 Å being introduced to push the target residue pairs towards the well when they are a long distance apart (Fig. S17). Here, the gradient width (d_b) of the contact well is the only free parameter of the 3G potential which depends on the protein size and determines the convergence speed and satisfaction rate of the contact-maps in combination with the inherent QUARK potential. As shown in Table S15, d_b is typically narrow, e.g., 6 Å, when the length of the target is relatively small, e.g., < 100. On the other hand, the gradient width increases to 12 Å when the length is >200, since simulations with larger size proteins are more difficult to converge and C-QUARK needs to use a wider well to draw the candidate residue pairs that are further apart in distance to the well smoothly and bring the residues pairs within 8 Å quickly. It is important that Eq. 1 is designed in a way that the potential curve is continuous and smooth (with $\partial E/\partial d = 0$) at all three transition points of $d_{ij} = 8, D$ and 80 Å, so that the contact restraints can be implemented smoothly without singularities. Furthermore, since contact prediction can only tell whether the distance between a residue pair $i-j$ is below 8 Å or not, we designed the 3G potential as a constant when the distance is < 8 Å. As almost all of the residue-residue distances in a normal size protein are lower than 80 Å, the potential is also designed as flat beyond the maximal distance threshold (80 Å). However, between 8 Å and 80 Å, we set the potential as two regions split at the transition point ($d_{ij} = D$). In the region above D , a relatively weaker force is used to avoid structural overpacking due to false positive contact predictions, while in the region below D , a stronger force is used to push contact restraints quickly satisfied since in this region the contact accuracy of the target residue pairs is supposed to be higher than that in the longer-distance regions (-because most of the adjacent residue pairs in the structure decoys are supposed to be more consistent with the inherent QUARK potential after the equilibrium of Monte Carlo simulations). Two trigonometric function style potentials are selected in the two regions to connect the flat areas, since trigonometric functions are simple, continuous, smooth, and differentiable.

10. The Reviewer commented:

9. The authors stated that besides 3G contact potential, another key contribution to the success of C-QUARK is the effective fragment assembly simulations. Is there any significant difference between the fragment assembly strategies used by QUARK and C-QUARK?

Thank you for the question, where the simple answer to the question is 'no', as the core part of the inherent fragment assembly strategies, including the major energy terms and REMC simulations, is largely the same between QUARK and C-QUARK. However, there are indeed some important differences between the two programs, in addition to the 3G contact potential. The most important change is the reparameterization of the force field to rebalance the contact restraints and the inherent QUARK potential based on the 243 training proteins. For instance, the weight (w_7) of the distance-profile energy term (E_{dp} in Eq. S2) was increased from 0.60 to 3.00 in the C-QUARK force field to allow the fragment-based potential to help filter out false positive contacts. Furthermore, we added a new energy term, which accounts for the distance between adjacent C α atoms ($E_{c\alpha}$ in Eq. S2 and Eq. S3), to penalize adjacent residue pair with C α -C α distances > 4Å. This term is

specifically designed to penalize broken backbones caused by fragment movements, as we have seen a stronger trend of chain breaking after introducing the strong contact restraints.

We added the following paragraph to discuss these changes (Page 12):

Besides the newly developed contact energy term (3G potential), the other energy terms have also been adjusted to maximize the folding performance of the 243 training proteins. For instance, the weight (w_7) of the distance-profile energy term (E_{dp} in Eq. S2) was increased from 0.60 to 3.00 in the C-QUARK force field to allow the fragment-based potential to help filter out false positive contacts. Furthermore, we added a new energy term, which accounts for the distance between adjacent C α atoms (E_{ca} in Eq. S2 and Eq. S3), to penalize adjacent residue pair with C α -C α distances $> 4\text{\AA}$. This term is specifically designed to penalize backbone breaks that can occur after fragment movements, as a stronger trend of bond-breaking was seen after the introduction of contact predictions in C-QUARK.

11. The Reviewer commented:

Minor comments:

1. Why only single-domain proteins are chosen? Can C-QUARK build structures for multi-domain proteins? Further experiments on these proteins will be very instructive.

Thank you for raising this important question. The major reason for the selection of single-domain proteins in the original version was the follow-up of the convention of the CASP experiment in which all the tertiary structure modeling results have been assessed at the domain-level, although many of the CASP targets contain multiple domains. This in fact reflects the relatively low ability of the field in modeling multi-domain proteins, probably because (1) multi-domain proteins have additional degree of freedom in domain orientation and therefore is difficult to model and (2) most of the current methods, including QUARK/C-QUARK, have been optimized for folding single-domain structures.

Nevertheless, following the Reviewer's suggestion, we selected 21 multi-domain proteins from CASP13, which contains in total 62 individual domains, for benchmarking the quality of multi-domain and single-domain modeling of C-QUARK. The result was summarized in **Table S10** and **Fig. S14**, and discussed in page 10 of the **Main Text**:

Finally, modeling multi-domain proteins is much harder than folding single-domain structures because of the introduction of additional degree of freedom in inter-domain orientations. For instance, the average TM-score (0.47) of the full-length models predicted by C-QUARK for the 21 multi-domain targets in CASP13 was much lower than that (0.65) for the individual domains (**Fig. S14** and **Table S10**). This is mainly due to the low accuracy of inter-domain contact prediction compared to intra-domain contact prediction, where the low contact accuracy is probably originated from the less-well constructed MSAs for the multi-domain sequences. Meanwhile, many energy terms of C-QUARK force field, including solvation and radius of gyration, have been designed and optimized for single-domain structure folding.

Table S10: Performance of C-QUARK on 21 CASP13 multi-domain proteins. The “Target” column is the name of each target. The second and third columns are the number of domains and the domain boundaries given by the CASP13 assessors for each target. The fourth column shows the TM-score of the C-QUARK first models for the full-length targets, and the fifth column shows the TM-scores of the C-QUARK first model for each individual domain of the targets. The last column is the average TM-score of the individual domains for each target.

Target	No. of domains	domain	TM-score of full-length model	TM-scores of domain models	Average TM-score of domain models
T0953s2	3	2-45;46-151,229-249; 152-228;	0.459	0.361,0.465, 0.286	0.371
T0957s1	2	2-37,92-163;38-91;	0.385	0.397,0.378	0.388
T0960	5	11-42;43-126; 127-215;216-279; 280-384;	0.289	0.158,0.437, 0.765,0.236, 0.714	0.462
T0963	5	9-39;40-121; 122-214;215-278; 279-372;	0.251	0.160,0.459, 0.773,0.250, 0.786	0.486
T0976	2	9-128;129-252;	0.703	0.839,0.830	0.835
T0977	2	59-359;360-563;	0.630	0.899,0.777	0.838
T0981	5	34-119;120-190,394-402; 191-393;403-513; 514-640;	0.317	0.535,0.251, 0.669,0.591, 0.640	0.537
T0982	2	11-145;146-277;	0.484	0.867,0.590	0.729
T0984	2	39-406,565-700;417-563;	0.863	0.875,0.789	0.832
T0987	2	11-195;196-402;	0.380	0.581,0.438	0.510
T0989	2	1-134;135-246;	0.364	0.477,0.313	0.395
T0990	3	1-76;77-134,348-520; 135-347;	0.241	0.577,0.371, 0.223	0.390
T0996	6	17-123;124-250; 251-350;351-483; 484-604;605-708; 15-400;401-853;	0.350	0.779,0.820, 0.816,0.750, 0.737,0.865	0.795
T0999	5	866-1045;1046-1289; 1290-1577;	0.429	0.986,0.769, 0.792,0.964, 0.890	0.880
T1000	2	10-92;93-523;	0.711	0.948,0.851	0.900
T1002	3	1-59;60-118; 127-270;	0.469	0.782,0.802, 0.777	0.787
T1004	3	66-151;152-228; 229-458;	0.558	0.793,0.673, 0.926	0.797
T1011	2	55-268,433-520;271-430;	0.580	0.792,0.874	0.833
T1014	2	1-159;160-276;	0.528	0.897,0.800	0.849
T1021s3	2	4-181;195-295;	0.426	0.636,0.452	0.544
T1022s1	2	2-157;158-224;	0.435	0.555,0.632	0.594

Figure S14: Boxplot and distribution of TM-scores for the first models produced by C-QUARK on 21 CASP13 multi-domain targets and the corresponding 62 individual domains.

Reviewers' Comments:

Reviewer #1:

Remarks to the Author:

The authors clearly answered to the questions I raised and edited the manuscript accordingly

Reviewer #2:

Remarks to the Author:

All of my comments have been addressed. I have no further comments.

Response to Reviewer #1

1. The Reviewer commented:

The authors clearly answered to the questions I raised and edited the manuscript accordingly.

We appreciate that the Reviewer satisfied with our revised manuscript.

Response to Reviewer #2

1. The Reviewer commented:

All of my comments have been addressed. I have no further comments.

We are glad to hear that the Reviewer satisfied with our revised manuscript.